# Biobank-scale genetic characterization of Alzheimer's disease and related dementias across diverse ancestries

Marzieh Khani[1], Fulya Akçimen [2,6], Spencer M. Grant[1,6], Suleyman Can Akerman [3,4], Paul Suhwan Lee [1], Faraz Faghri[1,5], Hampton Leonard[1,5], Jonggeol Jeffrey Kim [1], Mary B. Makarious [1,5], Mathew J. Koretsky[1,5], Jeffrey D. Rothstein[3,4], Cornelis Blauwendraat [1,2], Mike A. Nalls [1,5], Andrew Singleton[1] & Sara Bandres-Ciga [1] ✉

Alzheimer's disease and related dementias (AD/ADRDs) pose a significant global public health challenge. To effectively implement personalized therapeutic interventions on a global scale, it is essential to identify disease-causing, risk, and resilience factors across diverse ancestral backgrounds. This study leveraged biobank-scale data to conduct a large multi-ancestry whole-genome sequencing characterization of AD/ADRDs. We thoroughly explored the role of protein-coding and splicing variants from key genes associated with AD/ADRDs across 11 ancestries, utilizing data from five distinct biobanks, including a total of 25,001 cases and 93,542 controls. We compiled the most extensive catalog of known and novel genetic variation in AD/ADRDs in a global context, providing clinical insights into their genetic-phenotypic correlations. A thorough assessment of *APOE* revealed ancestry-driven modulation of *APOE*-associated AD/ADRDs, as well as disease-modifying effects conferred by several variants among *APOE* ε4 carriers. Finally, we present an accessible and user-friendly platform to support future ADRD research (https://niacard.shinyapps.io/MAMBARD_browser/).

In 2023, the World Health Organization reported that dementia affects ~55 million people worldwide[1]. This number is expected to reach ~152.8 million (ranging from 130.8 to 175.9 million) by 2050[2], placing a significant burden on healthcare infrastructure. Alzheimer's disease (AD), the most common form of dementia, represents roughly 60-70% of all cases[1]. Less prevalent forms, such as dementia with Lewy bodies (DLB) and frontotemporal dementia (FTD), each account for 10–15% of dementia cases[3,4].

Most of the research conducted thus far on the genetic underpinnings of dementia has primarily focused on populations of European ancestry, limiting the generalizability of findings[5]. Growing evidence indicates significant differences in the genetic architecture of disease among diverse ancestral populations, which raises concerns about the development of therapeutic interventions based on genetic targets primarily identified in a single population. Expanding research to include diverse ancestries is crucial for precision therapeutics. In the new era of personalized medicine, achieving accurate and effective disease-modifying treatments requires a comprehensive understanding of these diseases in a global context.

[1]Center for Alzheimer's and Related Dementias (CARD), National Institute on Aging and National Institute of Neurological Disorders and Stroke, National Institutes of Health, Bethesda, MD, USA. [2]Molecular Genetics Section, Laboratory of Neurogenetics, National Institute on Aging, National Institutes of Health, Bethesda, MD, USA. [3]Brain Science Institute, Johns Hopkins University School of Medicine, Baltimore, MD, USA. [4]Department of Neurology, Johns Hopkins University School of Medicine, Baltimore, MD, USA. [5]Data Tecnica LLC, Washington, DC, USA. [6]These authors contributed equally: Fulya Akçimen, Spencer M. Grant. ✉e-mail: sara.bandresciga@nih.gov

In recent years, researchers and healthcare institutions worldwide have undertaken ambitious efforts to create large-scale datasets encompassing diverse genetic ancestries, providing valuable insights into the genetic, environmental, and clinical factors influencing disease susceptibility and progression[6,7]. While more work remains in generating diverse genetic datasets that are dementia-specific, existing efforts can provide valuable insights into dementia research. Currently, All of Us (AoU), UK Biobank (UKB), 100,000 Genomes Project (100KGP), Alzheimer's Disease Sequencing Project (ADSP), and the Accelerating Medicines Partnership in Parkinson's Disease (AMP PD) are among the largest and most prominent publicly available dementia datasets worldwide.

A priority in elucidating the etiology of AD and related dementias (AD/ADRDs) lies in defining cumulative risk or in identifying highly penetrant disease-causing mutations; however, very little is known about genetic factors that enhance resistance to or protect against dementia. In genetics, protective variants reduce the risk of developing dementia or delay its onset. They confer protection via a loss-of-function or gain-of-function mechanism and can influence various biological pathways associated with the disease. Resilience variants (disease-modifying factors reducing the penetrance of risk loci) influence the development and course of the disease in individuals already at risk, potentially delaying symptom onset or reducing disease severity by interacting with pre-existing risk variants (genetic modifiers). To the best of our knowledge, 11 protective and 10 resilience variants have been reported in AD, with previous reports primarily focusing on the role of genetic variation in modulating AD risk among homozygous or heterozygous *APOE* ε4 carriers[8–14]. Understanding factors that confer protection or resilience can inform therapeutic strategies to reduce the overall burden of dementia, potentially decreasing healthcare costs and the societal impact of the disease.

In this study, we aimed to conduct a large and comprehensive multi-ancestry whole-genome sequencing characterization of AD/ADRDs potential disease-causing variation, as well as risk, protective, and disease-modifying factors, leveraging biobank-scale data. We screened genetic variants in key genes linked to these conditions, including *APP, PSEN1, PSEN2, TREM2, MAPT, GRN, GBA1, SNCA, TBK1, TARDBP,* and *APOE* across a total of 25,001 AD/ADRDs cases and 93,542 control individuals, collectively representing 11 ancestries. Furthermore, we assessed protective and disease-modifying variants among different *APOE* genotype carriers in those ancestry groups. Polygenic risk score (PRS) and burden analysis were performed in the most well-powered dataset (ADSP). This research is particularly relevant in the context of population-specific target prioritization for therapeutic interventions. Such advancements are crucial, as drug mechanisms supported by genetic insights have a 2.6 times higher likelihood of success than those without such support, underscoring the importance of including diverse genetic data to enhance therapeutic outcomes[15]. Here, we present genetic-phenotypic correlations among identified variants across all datasets and develop a user-friendly platform for the scientific community to help inform and support basic, translational, and clinical research on these debilitating conditions (https://niacard.shinyapps.io/MAMBARD_browser/). Figure 1 displays the demographic characteristics of cohorts under study. Figure 2 shows a summary of our workflow, which we explain in further detail below.

## Results
### Large-scale genetic characterization nominates known and novel potential disease-causing variants associated with Alzheimer's disease and related dementias
We used three different datasets (AoU, UKB, and 100KGP) to identify known and novel potential disease-causing variants. A summary of the variants that were identified can be found in Fig. 3 and Supplementary Fig. 1. Within the AoU dataset (discovery phase), we identified a total of

193 variants in 11 genes (*APP, PSEN1, PSEN2, TREM2, GRN, MAPT, GBA1, SNCA, TBK1, TARDBP,* and *APOE*). All variants and their allele frequencies in cases and controls across different ancestries are available in Supplementary Data 1. Among these variants, 36 were present only in cases and had a Combined Annotation Dependent Depletion (CADD) score > 20 (CADD score > 20 means that the variant is among the top 1% most pathogenic in the genome, as a proxy for its deleteriousness). All 36 identified variants were heterozygous. Of these, five were previously reported in AD or FTD (Table 1), while 31 were novel (Table 2). Of the five known variants, three were found in cases of European ancestry, one in cases of African ancestry, and one in cases of American Admixed ancestry. Among the 31 novel variants, 20 were found in cases of European ancestry, four in cases of African ancestry, two in cases of American ancestry, two in a case of Ashkenazi Jewish ancestry, and three in cases of African Admixed ancestry.

Within the UKB (discovery phase), we identified a total of 815 variants in the *APP, PSEN1, PSEN2, TREM2, GRN, MAPT, GBA1, SNCA, TBK1, TARDBP,* and *APOE* genes (Supplementary Data 2). Among these, 121 variants were present only in cases and had a CADD score > 20. All 121 identified variants were heterozygous. Of these, 20 were previously reported as disease-causing in AD, FTD, frontotemporal lobar degeneration (FTLD), and Gaucher disease (Table 1), while 101 were novel (Table 2). A majority (*n* = 113) of the variants were identified in individuals of European genetic ancestry, while three were identified in cases of African ancestry, three in cases of South Asian ancestry, and one each in cases of East Asian and Ashkenazi Jewish ancestries. The allele frequencies of the variants across different ancestries are reported in Supplementary Data 2.

Within the 100KGP data (discovery phase), we identified a total of 13 variants in the *APP, PSEN1, PSEN2, GRN, GBA1, TBK1,* and *TARDBP* genes (Supplementary Data 3). Among cases, no variants were identified in the *MAPT, TREM2, SNCA,* and *APOE* genes. Of the 13 variants, four were only present in cases and had a CADD score >20. All four identified variants were heterozygous and previously reported in individuals of European ancestry. Among these four variants, *PSEN1* p.R269H and *TARDBP* p.G287S had been previously reported as causes of AD and amyotrophic lateral sclerosis (ALS), respectively (Table 1), while the remaining two variants in the *APP* and *PSEN2* genes were novel (Table 2). The allele frequency of each variant is presented in Supplementary Data 3.

We next used the ADSP and AMP PD cohorts to replicate the disease-associated variant findings listed above.

### Replication analyses support the relevance of identified genetic variation across diverse ancestries
Nine variants identified in AoU, 20 identified in UKB, and four identified in 100KGP were replicated in AD cases in the ADSP cohort (Supplementary Data 4). Below we detail these variants and their occurrences in both European and non-European participants.

Among the nine variants found in AoU that were replicated in ADSP, three variants—*APP* p.A713T, *PSEN1* p.A79V, and *TARDBP* p.G287S — had been previously reported, while six variants—*APP* p.L597W, *MAPT* p.G701R, *MAPT* p.G750S, *SNCA* p.Q99R, *TBK1* p.N42S, and *TBK1* p.R507I—were novel. The *APP* p.L597W variant was found in African, African Admixed, and Complex Admixture History ancestries. Searching for other dementia cases resulted in the identification of *APP* p.A713T in one DLB case and *PSEN1* p.A79V in a possible AD case according to ADSP diagnosis criteria. The allele frequency of each variant per genetic ancestry in cases and controls is reported in Supplementary Data 4.

Among the four variants identified in the 100KGP dataset that were replicated in the ADSP cohort, *PSEN1* p.R269H and *TARDBP* p.G287S were previously reported, while *PSEN2* p.D320N and *APP* p.Y407H were novel. We observed the previously reported *PSEN1* p.R269H variant in five cases and no control participants. This variant

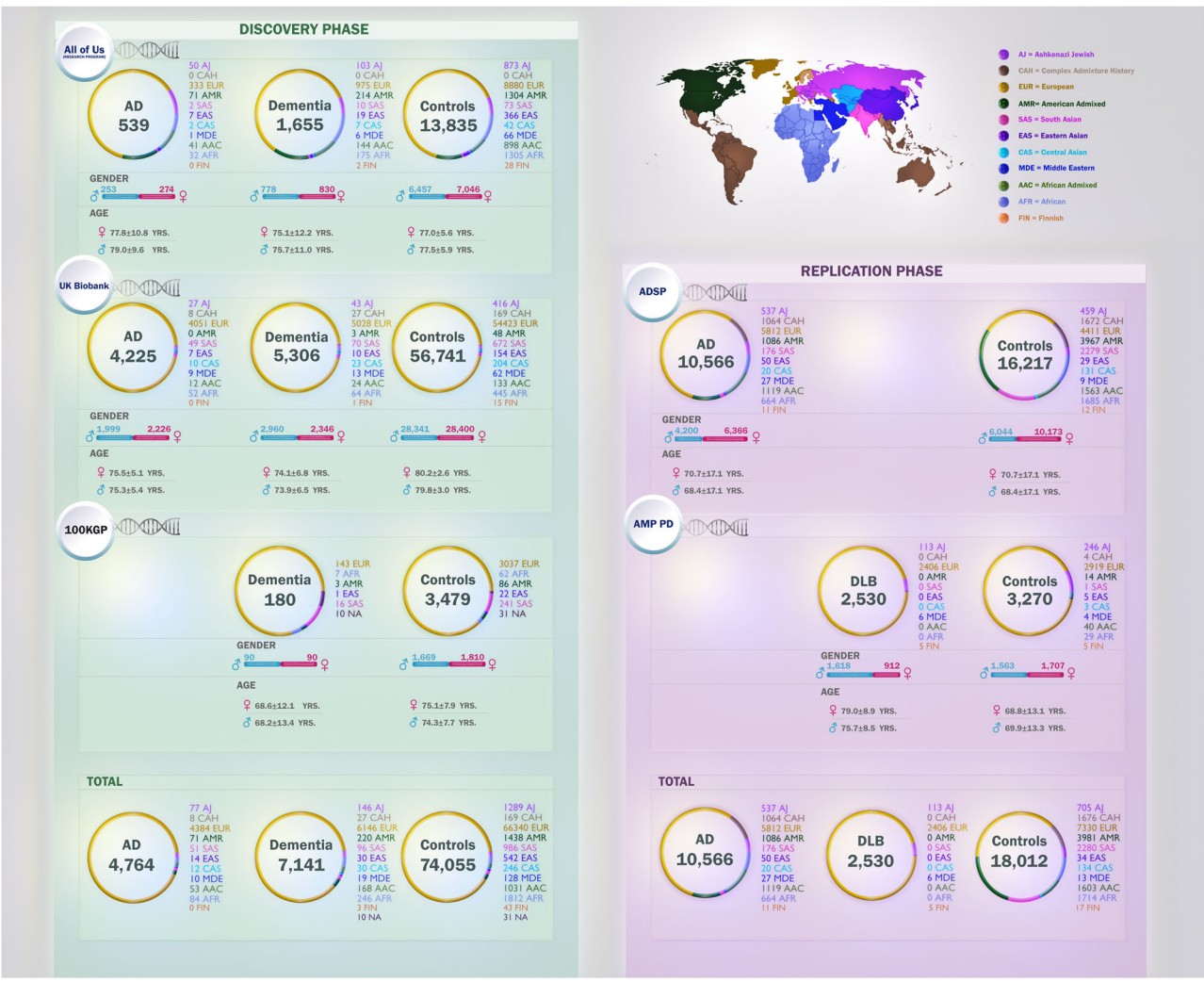

**Fig. 1 | Demographic and clinical characteristics of biobank-scale cohorts under study.** The figure illustrates distributions of age, sex, and the number of cases and controls per ancestry across five datasets in this study: All of Us (AoU), Alzheimer's Disease Sequencing Project (ADSP), 100,000 Genomes Project (100KGP), UK Biobank (UKB), and Accelerating Medicines Partnership in Parkinson's Disease (AMP PD). Ancestries represented include European (EUR), African (AFR), American Admixed (AMR), African Admixed (AAC), Ashkenazi Jewish (AJ), Central Asian (CAS), Eastern Asian (EAS), South Asian (SAS), Middle Eastern (MDE), Finnish (FIN), and Complex Admixture History (CAH).

was found in European ancestry individuals from the 100KGP cohort and was also observed in individuals of African Admixed ancestry (two cases) and European ancestry (three cases) in the ADSP dataset. *TARDBP* p.G287S was found in one European control individual and absent in cases. Of the novel variants, the *PSEN2* p.D320N variant was found in five controls and was not observed in any cases, while the *APP* p.Y407H variant was observed in two cases and one control. Searching for additional cases led to the discovery of *PSEN1* p.R269H in a patient with mild cognitive impairment (MCI).

Among the 20 variants identified in the UKB cohort that were replicated in the ADSP dataset, six variants—*PSEN1* p.A79V (five cases and one control), *PSEN1* p.R269H (five cases), *GRN* p.R493X (three cases), *MAPT* p.R406W (four cases and one control), *GBA1* p.W223R (one case and one control), and *TBK1* p.R228H (one case)—have been previously reported. The remaining 14 variants were novel. Most of the novel variants were found in European cases in the UKB. Among the known variants, the *PSEN1* p.R269H variant was found in cases of both African Admixed and European ancestries, and *GBA1* p.W223R was found in a case of Complex Admixture History ancestry and a control of African ancestry. The four remaining known variants were observed in individuals of European ancestry in the ADSP cohort.

Novel variants identified in non-European participants include *PSEN1* p.R220Q (one African case, two European cases, and one African Admixed control), *PSEN1* p.T291A (one African Admixed case, one American Admixed control, and two controls with Complex Admixture History), *MAPT* p.A60G (two African Admixed cases, one African Admixed control, two African controls, and four controls with Complex Admixture History), *GRN* p.V490M (one African Admixed case), *MAPT* p.R103W (one American Admixed control), *APP* p.P251S (one South Asian control), *APOE* p.Q65X (one African control), and *APOE* p.E275G (one African Admixed control). Searching for other cases resulted in the identification of *PSEN1* p.A79V in one possible AD patient, *PSEN1* p.R269H and *MAPT* p.A60G in two independent MCI patients, and *GBA1* p.W223R and *APP* p.A209T in two independent progressive supranuclear palsy (PSP) patients (Supplementary Data 4).

We identified a novel *SNCA* variant (p.Q99R) in the AoU dataset, while the UKB dataset revealed two additional variants in *SNCA*: p.P90H and p.A91S. Both p.P90H and p.A91S were predicted to be likely pathogenic according to prediction estimates and have not been previously reported as disease-causing. Notably, the *SNCA* p.Q99R variant was replicated in the ADSP cohort. All three variants were heterozygous, and none of these variants were found in any controls

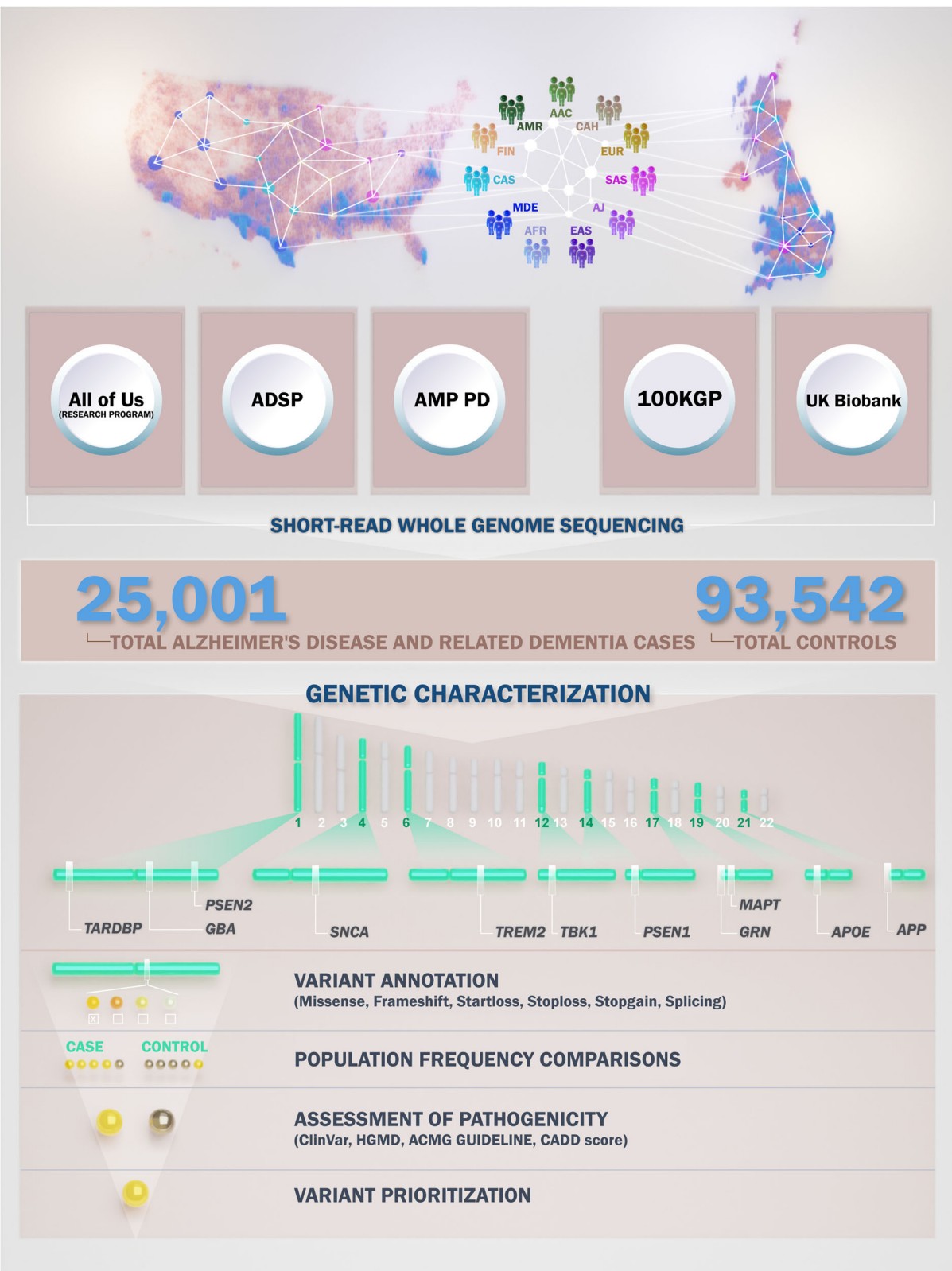

**Fig. 2 | Workflow.** Our workflow began with creating cohorts within the datasets. We leveraged short-read whole genome sequencing data to characterize genes of interest. Variant annotation focused on missense, frameshift, start loss, stop loss, stop gain, and splicing variants. Next, we compared the frequency of identified variants in cases and controls. Pathogenicity assessment involved using ClinVar, Human Gene Mutation Database (HGMD), American College of Medical Genetics and Genomics (ACMG) guidelines, and Combined Annotation Dependent Deple-tion (CADD) scores. Finally, we prioritized variants that were present only in the case cohort and had a CADD score greater than 20.

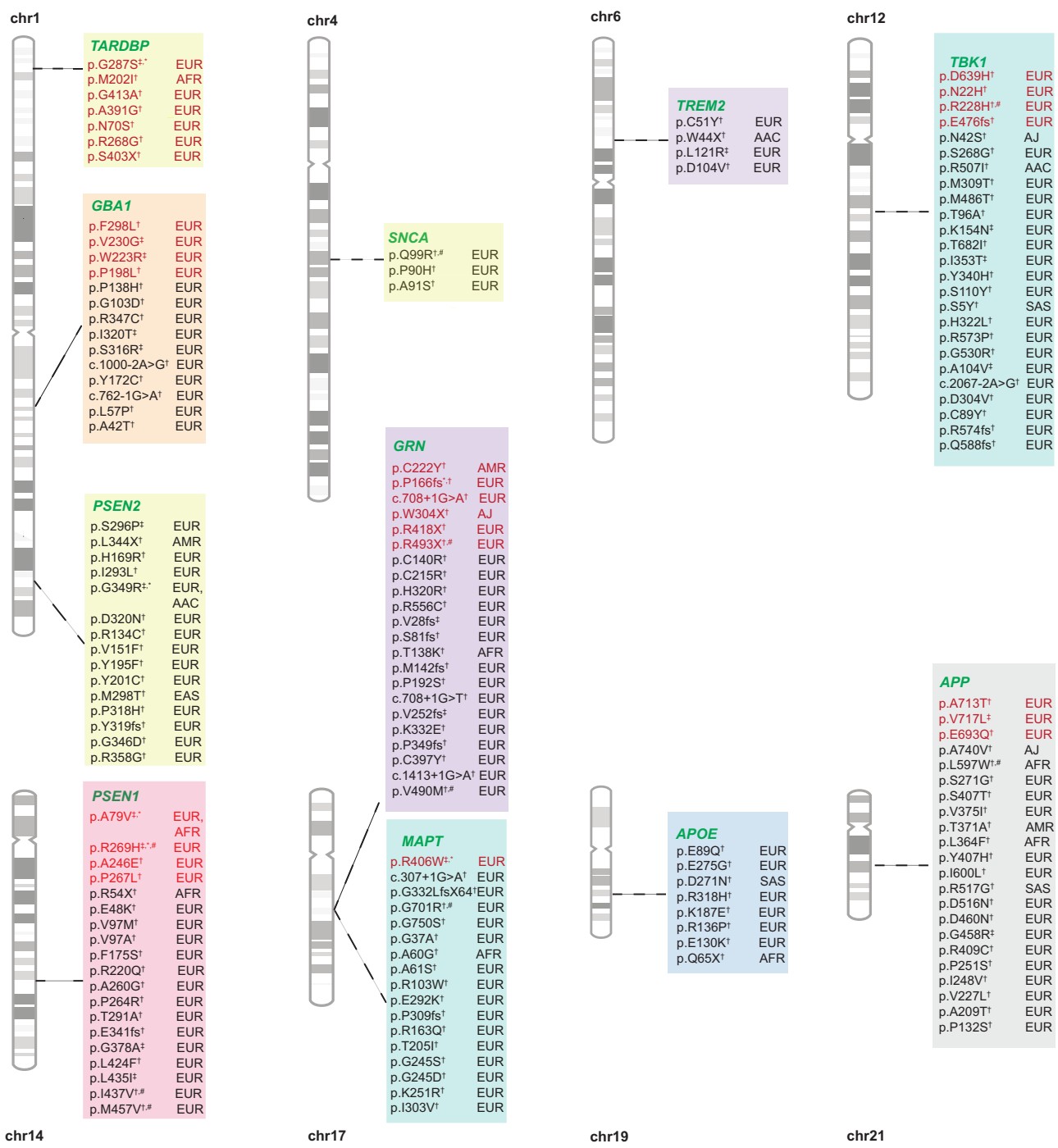

**Fig. 3 | Summary of known and novel variants identified during the discovery phase.** Variants in red are known, while variants in black are novel. † Singleton variants identified in the discovery phase; ‡ Variants found in more than one individual during the discovery phase; * Variants replicated across biobanks in the discovery phase; # Variants replicated only in cases during the replication phase across biobanks.

across these datasets. However, the age at onset (AAO) of these variant carriers is not consistent with a potential disease-causing highly deleterious effect.

Our analyses of multiple datasets identified 12 variants (three from AoU, eight from UKB, and one from 100KGP) that were absent in the ADSP control cohort.

Across each of our discovery datasets, we identified six candidate variants—*APP* p.A713T, *MAPT* p.G750S, *GRN* p.V490M, *GRN* p.R493X, *APP* p.D516N, and *TARDBP* p.G287S—present in AMP PD (DLB cases and controls). *GRN* p.V490M and *TARDBP* p.G287S were present in one

control and absent in cases, *GRN* p.R493X was present in one case and absent in controls, while the other three were present across both cases and controls. The allele frequencies of these variants are detailed in Supplementary Data 5.

Among the 156 variants identified in this study, 18 were found exclusively in non-European ancestries. Notably, *APP* p.L597W and *MAPT* p.A60G were replicated in African and African Admixed ancestries across different datasets. These data highlight the potential significance of these variants in groups that are often underrepresented in genomic studies.

**Table 1 | Discovery phase: Multi-ancestry summary of known potential disease-causing variants only present in Alzheimer's disease and related dementia cases in AoU, 100KGP and UKB**

| | Gene | Position | rs ID/ClinVar ID | cDNA changes | Protein change/Splicing | Clinical significance | HGMD/Disease reported | CADD | Genetic ancestry | Zygosity | GnomAD |
|---|---|---|---|---|---|---|---|---|---|---|---|
| **AoU** | | | | | | | | | | | |
| | APP | chr21: 25891796[a] | rs63750066 | C>T | p.A713T | Pathogenic, Likely pathogenic, VUS | CM930033, AD | 26.9 | EUR | Het | 3.64E-05 |
| | **PSEN1** | **chr14:73170945** | **rs63749824** | **C>T** | **p.A79V** | **Pathogenic, Likely pathogenic** | **CM981649, AD** | **26.1** | **AFR** | **Het** | **1.34E-05** |
| | **MAPT** | **chr17:46024061** | **rs63750424** | **C>T** | **p.R406W** | **Pathogenic,VUS** | **CM981237, FTD with parkinsonism** | **23.9** | **EUR** | **Het** | **1.98E-05** |
| | GRN | chr17:44350757 | rs77770477 | G>A | p.C222Y | VUS | CM149714, AD | 28.9 | AMR | Het | 2.24E-05 |
| | **TARDBP** | **chr1:11022268[a]** | **rs80356719** | **G>A** | **p.G287S** | **Pathogenic, Likely pathogenic, VUS** | **CM081839, ALS** | **22.6** | **EUR** | **Het** | **4.32E-05** |
| **100KGP** | **PSEN1** | **chr14:73198067** | **rs63750900** | **G>A** | **p.R269H** | **Pathogenic, Likely pathogenic** | **CM971254, AD** | **29.9** | **EUR** | **Het** | **8.48E-06** |
| | **TARDBP** | **chr1:11022268[a]** | **rs80356719** | **G>A** | **p.G287S** | **Pathogenic, Likely pathogenic, VUS** | **CM081839, ALS** | **22.6** | **EUR** | **Het** | **4.32E-05** |
| **UKB** | GBA1 | chr1:155237446 | rs1671825414 | G>T | p.F298L | Likely pathogenic | CM000164, Gaucher disease 2 | 23.6 | EUR | Het | 7.20E-06 |
| | GBA1 | chr1:155238206 | rs381427 | A>C | p.V230G | Pathogenic/VUS | CM980833, Gaucher disease | 22.5 | EUR | Het | 3.39E-06 |
| | GBA1 | chr1:155238228 | rs61748906 | A>G | p.W223R | Pathogenic/Likely pathogenic/VUS | CM001166, Gaucher disease 2 | 28 | EUR | Het | 1.19E-05 |
| | GBA1 | chr1:155238302 | rs80222298 | G>A | p.P198L | Likely pathogenic, VUS | CM980827, Gaucher disease | 28.6 | EUR | Het | 8.62E-07 |
| | **PSEN1** | **chr14:73170945** | **rs63749824** | **C>T** | **p.A79V** | **Pathogenic, Likely pathogenic** | **CM981649, AD** | **26.1** | **EUR** | **Het** | **1.10E-05** |
| | PSEN1 | chr14:73192832 | rs63750526 | C>A | p.A246E | Pathogenic, Likely pathogenic | CM951075, AD | 25.4 | EUR | Het | 9.00E-07 |
| | PSEN1 | chr14:73198061 | rs63750779 | C>T | p.P267L | Likely pathogenic | CM033803, AD | 25.8 | EUR | Het | 9.01E-07 |
| | **PSEN1** | **chr14:73198067** | **rs63750900** | **G>A** | **p.R269H** | **Pathogenic, Likely pathogenic** | **CM971254, AD** | **29.9** | **EUR** | **Het** | **8.48E-06** |
| | GRN | chr17:44350449 | VCV001922048.3 | ->CTGTGAAGACAGGGTGCACTGCTGT | p.P166fs[a] | Pathogenic | Not reported, FTD | 34 | EUR | Het | 8.48E-07 |
| | GRN | chr17:44350801 | rs63749817 | G>A | c.708+1G>A | Pathogenic/Likely pathogenic | CS200794, FTD | 34 | EUR | Het | 3.42E-06 |
| | GRN | chr17:44351438 | rs63751177 | G>A | p.W304X | Pathogenic | CM064045, FTD - CM188618, FTLD | 39 | AJ | Het | 0 |
| | GRN | chr17:44352087 | rs63751180 | C>T | p.R418X | Pathogenic | CM062773, FTD | 25.6 | EUR | Het | 4.50E-06 |
| | GRN | chr17:44352404 | rs63751294 | C>T | p.R493X | Pathogenic | CM064044, FTD | 36 | EUR | Het | 1.62E-05 |
| | **MAPT** | **chr17:46024061** | **rs63750424** | **C>T** | **p.R406W** | **Pathogenic, VUS** | **CM981237, FTD with parkinsonism** | **23.9** | **EUR** | **Het** | **1.98E-05** |
| | APP | chr21:25891784 | rs63750264 | C>T | p.V717L | Pathogenic/Likely pathogenic | CM003587, AD | 26.8 | EUR | Het | 1.80E-06 |
| | APP | chr21:25891856 | rs63750579 | C>G | p.E693Q | Pathogenic/Likely pathogenic | CM920067, AD | 27.1 | EUR | Het | - |
| | TBK1 | chr12:64497215 | rs778577820 | G>C | p.D639H | VUS | CM187872, AD | 22.9 | EUR | Het | 1.81E-06 |

**Table 1 (continued) | Discovery phase: Multi-ancestry summary of known potential disease-causing variants only present in Alzheimer's disease and related dementia cases in AoU, 100KGP and UKB**

| Gene | Position | rs ID/ClinVar ID | cDNA changes | Protein change/ Splicing | Clinical significance | HGMD/Disease reported | CADD | Genetic ancestry | Zygosity | GnomAD |
|---|---|---|---|---|---|---|---|---|---|---|
| *TBK1* | chr12:64455934 | rs576726084 | A>C | p.N22H | VUS | CM152626, ALS | 26.4 | EUR | Het | 8.48E-06 |
| *TBK1* | chr12:64474372 | rs748622208 | G>A | p.R228H | VUS | CM152640, ALS | 34 | EUR | Het | 7.63E-06 |
| *TBK1* | chr12:64488573 | rs1131690783 | A>- | p.E476fs | - | CD171626, MND | 34 | EUR | Het | 2.71E-06 |

*AoU All of Us, 100KGP 100,000 Genomes Project, UKB UK Biobank, EUR European, AFR African, AMR American Admixed, AAC African Admixed, AJ Ashkenazi Jewish, CAS Central Asian, EAS Eastern Asian, SAS South Asian, MDE Middle Eastern, FIN Finnish, CAH Complex Admixture History, HGMD Human Gene Mutation Database. The frequency of gnomAD refers to the frequency in the ancestry where the variations were found. Bold variants were replicated in different databases. Disease reported refers to the disease for which the variants were previously reported. Position refers to GRCh 38.*
*cDNA complementary DNA, VUS variant uncertain significance, Het Heterozygous, CADD Combined Annotation Dependent Depletion, AD Alzheimer's disease, FTD frontotemporal dementia, ALS Amyotrophic lateral sclerosis, MND Motor neuron disease.*
*[a]These variants were not replicated only in cases across the diverse biobanks in the discovery phase. Clinical significance based on dbSNP, ClinVar, and ACMG guideline.*

Supplementary Fig. 2 shows the allele frequencies of all identified known and novel variants with a CADD > 20 in the discovery and replication phases across all ancestries in each biobank.

### Previously reported disease-causing variants raise questions about potential pathogenicity

In addition to identifying potential disease-causing variants, we also identified variants in *SNCA*, *APP*, *TARDBP*, *GRN*, and *GBA1* that may not cause disease or may be risk factors of incomplete penetrance based on their presence in control participants.

Although the *SNCA* p.H50Q variant was initially identified as a disease-causing mutation in Parkinson's disease (PD)[16], subsequent research has challenged its pathogenicity[17]. Our study confirms that it is not pathogenic across other synucleinopathies such as DLB, based on its occurrence in five European controls in AoU and 28 European controls in UKB (Supplementary Data 6).

Additionally, several research studies have reported the *APP* p.A713T variant to be disease-causing[18,19]. In our study, we found this variant in heterozygous state in five control individuals: two in UKB, two in ADSP, and one in AMP PD. Interestingly, the *APP* p.E665D variant, which has been widely reported to cause AD[20,21], was found in one control in AoU in her late 70s. However, it is possible that the variant shows incomplete penetrance, that this individual may develop disease later in life, or that she may even harbor unidentified resilient genetic variation. Another previous study evaluating the role of *APP* p.E665D also questioned the pathogenicity of this variant[22].

*TARDBP* p.G287S, previously reported as disease-causing for ALS[23], was found in five controls—three in UKB, one in ADSP, and one in AMP PD. Similarly, *TARDBP* p.N390S, also previously reported as disease-causing for ALS[24], was found in two controls in UKB.

*GBA1* coding variants generally exhibit incomplete penetrance and act as genetic risk factors in heterozygous state. Homozygous *GBA1* variants, including the p.T75del and c.115+1G>A mutations, have been reported to cause Gaucher disease[25–28]. We found these two variants in heterozygous state in one case and one control in AoU. *GBA1* p.T75del was found in individuals of African ancestry, and *GBA1* c.115+1G>A was found in individuals of European ancestry in both a case and a control. The *GBA1* c.115+1G>A variant was also found in nine European controls in the UKB cohort. We identified 13 additional heterozygous variants in *GBA1* (p.R502C, p.A495P, p.L483R, p.D448H, p.E427X, p.G416S, p.N409S, p.R398X, p.R296Q, p.G241R, p.N227S, p.S212X, and p.R159W) that have been reported as disease-causing for Gaucher disease in homozygous state. Three variants in *GRN*, including two loss of function variants (p.Q130fs and p.Y294X) and one splicing variant (c.708+6_708+9del), have been previously reported to cause FTD, FTLD, and neurodegenerative disease[25,29–43]. Each of these 16 variants were found in several control individuals. The frequency and number of carriers for each variant in cases and controls are reported in Supplementary Data 6.

### Genetic-phenotypic correlations provide valuable clinical insights

Clinical data for the identified variants are summarized in Supplementary Data 7 and 8. Here, we briefly explain the main findings.

The *GRN* p.R493X variant is the most reported pathogenic mutation in this gene. This variant has been associated with several types of dementia, including FTD, FTLD, primary progressive aphasia, AD, and corticobasal degeneration. It is known to be more frequently identified among FTD cases, particularly in early-onset forms[44]. In one study investigating the genetics underlying disease etiology in 1118 DLB patients, this variant was reported in a single case, presenting a wide range of neurological phenotypes that could not lead to a conclusive diagnosis. Severe dementia, parkinsonism, and visual hallucinations suggested a clinical diagnosis of AD or mixed vascular dementia. However, the final neuropathological diagnosis was

**Table 2 | Discovery phase: Multi-ancestry summary of novel potential disease-causing variants only present in Alzheimer's disease and related dementia cases in AoU, 100KGP and UKB**

| Gene | Position | rs ID/ClinVar ID | cDNA changes | Protein change/ Splicing | Clinical significance | CADD | PP2 prediction | Genetic ancestry | Zygosity | GnomAD |
|---|---|---|---|---|---|---|---|---|---|---|
| **AoU** | | | | | | | | | | |
| APP | chr21:25881764 | Novel | G>A | p.A740V | VUS | 26.7 | Probably Damaging | AJ | Het | 0 |
| APP | chr21:25911860 | rs765301301 | A>C | p.L597W | VUS | 25.7 | Probably Damaging | AFR | Het | 1.47E-04 |
| APP | chr21:26021894 | Novel | T>C | p.S271G | VUS | 24.5 | Benign | EUR | Het | 0 |
| APP | chr21:25982349 | rs2042464223 | A>T | p.S407T | VUS | 24.2 | Probably Damaging | EUR | Het | 3.60E-06 |
| APP | chr21:25982445[a] | rs141331202 | C>T | p.V375I | VUS | 25.7 | Probably Damaging | EUR | Het | 3.22E-05 |
| APP | chr21:25982457 | rs747438691 | T>C | p.T371A | VUS | 24.6 | Possibly Damaging | AMR | Het | 4.47E-05 |
| APP | chr21:25997360 | rs749453173 | G>A | p.L364F | VUS | 23.9 | Benign | AFR | Het | 1.74E-04 |
| PSEN1 | chr14:73170869 | Novel | C>T | p.R54X | Pathogenic | 36 | - | AFR | Het | 0 |
| PSEN2 | chr1:226890133 | rs1410382029 | T>C | p.S296P | VUS | 33 | Probably Damaging | EUR | Het | 0 |
| PSEN2 | chr1:226891797 | Novel | CT>C | p.L344X | Pathogenic | 23.9 | - | AMR | Het | 0 |
| PSEN2 | chr1:226888098 | rs1661490243 | A>G | p.H169R | VUS | 26.2 | Probably Damaging | EUR | Het | 8.48E-07 |
| PSEN2 | chr1:226890124 | rs199689738 | A>T | p.I293L | VUS | 27.1 | Possibly Damaging | EUR | Het | 4.16E-05 |
| **PSEN2** | **chr1:226891817** | **rs759669954** | **G>A** | **p.G349R** | **VUS** | **22.1** | **Benign** | **AAC** | **Het** | **6.67E-05** |
| GRN | chr17:44350296 | rs1248058567 | T>C | p.C140R | VUS | 24.2 | Probably Damaging | EUR | Het | 1.80E-06 |
| GRN | chr17:44350735 | rs1201429668 | T>C | p.C215R | VUS | 29.5 | Probably Damaging | EUR | Het | 1.70E-06 |
| GRN | chr17:44351575 | Novel | A>G | p.H320R | VUS | 26.2 | Probably Damaging | EUR | Het | 8.99E-07 |
| GRN | chr17:44352682 | rs63750116 | C>T | p.R556C | VUS | 24.7 | Probably Damaging | EUR | Het | 7.19E-06 |
| MAPT | chr17:45974472 | rs747085337 | G>A | c.307+1G>A | - | 23.3 | - | EUR | Het | 6.33E-06 |
| MAPT | chr17:45983788 | Novel | AGGGGCCCCTGGAGAGGGGCCAGAGGCCC>A | p.G332LfsX64 | - | 27.3 | - | EUR | Het | 0 |
| MAPT | chr17:46018716 | rs948573449 | G>A | p.G701R | VUS | 34 | Probably Damaging | EUR | Het | 8.10E-06 |
| MAPT | chr17:46024088[a] | rs768841567 | G>A | p.G750S | VUS | 33 | Probably Damaging | EUR | Het | 2.20E-05 |
| TREM2 | chr6:41161502 | rs369181900 | C>T | p.C51Y | VUS | 27.9 | Probably Damaging | EUR | Het | 1.44E-05 |
| TREM2 | chr6:41161523 | Novel | C>T | p.W44X | Pathogenic | 37 | - | AAC | Het | 0 |
| GBA1 | chr1:155239657 | rs759174705 | G>T | p.P138H | VUS | 22.3 | Benign | EUR | Het | 2.54E-06 |
| GBA1 | chr1:155239762 | rs748485792 | C>T | p.G103D | VUS | 20.4 | Benign | EUR | Het | 3.60E-06 |
| SNCA | chr4:89822256 | rs757477802 | T>C | p.Q99R | VUS | 21.9 | Benign | EUR | Het | 1.36E-05 |
| TARDBP | chr1:11020491 | Novel | G>A | p.M202I | VUS | 22.2 | Benign | AFR | Het | - |

**Table 2 (continued) | Discovery phase: Multi-ancestry summary of novel potential disease-causing variants only present in Alzheimer's disease and related dementia cases in AoU, 100KGP and UKB**

| Gene | Position | rs ID/ClinVar ID | cDNA changes | Protein change/ Splicing | Clinical significance | CADD | PP2 prediction | Genetic ancestry | Zygosity | GnomAD |
|---|---|---|---|---|---|---|---|---|---|---|
| TARDBP | chr1:11022647 | Novel | G>C | p.G413A | Likely pathogenic | 23.2 | Probably Damaging | EUR | Het | - |
| TBK1 | chr12:64460226 | rs748061846 | A>G | p.N42S | Likely benign, VUS | 22.5 | Benign | AJ | Het | 0 |
| TBK1 | chr12:64480112 | rs746971642 | A>G | p.S268G | VUS | 21.5 | Possibly Damaging | EUR | Het | 1.80E-06 |
| TBK1 | chr12:64490118 | rs1251738886 | G>T | p.R507I | VUS | 25.7 | Probably Damaging | AAC | Het | 2.41E-05 |
| **100KGP** | | | | | | | | | | |
| APP | chr21:25954665[a] | rs779792929 | A>G | p.Y407H | VUS | 25.6 | Probably Damaging | EUR | Het | 8.31E-05 |
| PSEN2 | chr1:226891349 | rs565698726 | G>A | p.D320N | VUS | 21.8 | Benign | EUR | Het | 5.93E-06 |
| **UKB** | | | | | | | | | | |
| GBA1 | chr1:155235769 | rs747284798 | G>A | p.R347C | VUS | 31 | Probably damaging | EUR | Het | 8.47E-06 |
| GBA1 | chr1:155236249 | rs1057519358 | A>G | p.I320T | VUS | 25.3 | Probably damaging | EUR | Het | 8.10E-06 |
| GBA1 | chr1:155236262 | Novel | T>G | p.S316R | VUS | 27.7 | Probably damaging | EUR | Het | 2.70E-06 |
| GBA1 | chr1:155236471 | Novel | T>C | c.1000-2A>G | - | 31 | - | EUR | Het | - |
| GBA1 | chr1:155237564 | Novel | T>C | p.Y172C | VUS | 28 | Probably damaging | EUR | Het | 4.50E-06 |
| GBA1 | chr1:155237579 | Novel | G>A | c.762-1G>A | - | 26.8 | - | EUR | Het | - |
| GBA1 | chr1:155239639 | Novel | A>G | p.L57P | Likely pathogenic | 28.2 | Probably damaging | EUR | Het | 8.99E-07 |
| GBA1 | chr1:155239685 | rs1671971599 | C>T | p.A42T | VUS | 21.1 | Possibly damaging | EUR | Het | 0 |
| PSEN2 | chr1:226885581 | rs1363866270 | C>T | p.R134C | VUS | 32 | Probably damaging | EUR | Het | 1.80E-06 |
| PSEN2 | chr1:226885632 | Novel | G>T | p.V151F | VUS | 20.8 | Benign | EUR | Het | 8.99E-07 |
| PSEN2 | chr1:226888846 | Novel | A>T | p.Y195F | VUS | 28.2 | Possibly damaging | EUR | Het | 8.09E-06 |
| PSEN2 | chr1:226888864 | rs200410369 | A>G | p.Y201C | VUS | 28.5 | Probably damaging | EUR | Het | 1.27E-05 |
| PSEN2 | chr1:226891284 | rs1482790603 | T>C | p.M298T | VUS | 25.3 | Possibly damaging | EAS | Het | 1.34E-04 |
| PSEN2 | chr1:226891344 | Novel | C>A | p.P318H | VUS | 23.7 | Possibly damaging | EUR | Het | 1.80E-06 |
| PSEN2 | chr1:226891347 | Novel | A>- | p.Y319fs | - | 27.9 | - | EUR | Het | - |
| PSEN2 | chr1:226891809 | rs1365789341 | G>A | p.G346D | VUS | 20.3 | Benign | EUR | Het | 8.99E-07 |
| **PSEN2** | **chr1:226891817** | **rs759669954** | **G>A** | **p.G349R** | **VUS** | **22.1** | **Benign** | **EUR** | **Het** | **0** |
| PSEN2 | chr1:226891844 | Novel | A>G | p.R358G | VUS | 23.4 | Benign | EUR | Het | 9.00E-07 |
| PSEN1 | chr14:73170851 | rs1377702483 | G>A | p.E48K | VUS | 23.1 | Possibly damaging | EUR | Het | 2.54E-06 |

**Table 2 (continued) | Discovery phase: Multi-ancestry summary of novel potential disease-causing variants only present in Alzheimer's disease and related dementia cases in AoU, 100KGP and UKB**

| Gene | Position | rs ID/ClinVar ID | cDNA changes | Protein change/Splicing | Clinical significance | CADD | PP2 prediction | Genetic ancestry | Zygosity | GnomAD |
|---|---|---|---|---|---|---|---|---|---|---|
| PSEN1 | chr14:73170998 | rs63750852 | G>A | p.V97M | VUS | 28.3 | Probably damaging | EUR | Het | 7.63E-06 |
| PSEN1 | chr14:73170999 | rs1356498068 | T>C | p.V97A | VUS | 25.6 | Probably damaging | EUR | Het | 8.99E-07 |
| PSEN1 | chr14:73186896 | rs63750771 | T>C | p.F175S | VUS | 23.7 | Probably damaging | EUR | Het | 9.00E-07 |
| PSEN1 | chr14:73192754 | rs763831389 | G>A | p.R220Q | VUS | 23.8 | Possibly damaging | EUR | Het | 5.93E-06 |
| PSEN1 | chr14:73198040 | Novel | C>G | p.A260G | VUS | 26.8 | Probably damaging | EUR | Het | 9.11E-07 |
| PSEN1 | chr14:73198052 | Novel | C>G | p.P264R | Likely pathogenic | 25.7 | Probably damaging | EUR | Het | 9.02E-07 |
| PSEN1 | chr14:73206388 | rs63750298 | A>G | p.T291A | VUS | 27.8 | Possibly damaging | EUR | Het | 5.93E-06 |
| PSEN1 | chr14:73211836 | Novel | ->GCCC | p.E341fs | - | 34 | - | EUR | Het | - |
| PSEN1 | chr14:73217129 | rs63750323 | G>C | p.G378A | Likely pathogenic | 25.4 | Probably damaging | EUR | Het | 5.40E-06 |
| PSEN1 | chr14:73219155 | rs555358260 | C>T | p.L424F | Likely pathogenic | 25.1 | Probably damaging | EUR | Het | - |
| PSEN1 | chr14:73219188 | Novel | C>A | p.L435I | Likely pathogenic | 25.8 | Probably damaging | EUR | Het | 1.80E-06 |
| PSEN1 | chr14:73219194 | rs764971634 | A>G | p.I437V | Likely pathogenic, VUS | 23 | Benign | EUR | Het | 6.78E-06 |
| PSEN1 | chr14:73219254 | rs1430581353 | A>G | p.M457V | VUS | 23.5 | Probably damaging | EUR | Het | 1.80E-06 |
| GRN | chr17:44349248 | rs63751057 | ->GCCT | p.V28fs | - | 33 | - | EUR | Het | 4.50E-06 |
| GRN | chr17:44349529 | rs63751193 | C>- | p.S81fs | Pathogenic | 32 | - | EUR | Het | 8.99E-07 |
| GRN | chr17:44350291 | rs146769257 | C>A | p.T138K | VUS | 24.6 | Probably damaging | AFR | Het | 8.96E-05 |
| GRN | chr17:44350303 | Novel | ->GGTC | p.M142fs | - | 26.1 | - | EUR | Het | - |
| GRN | chr17:44350553 | Novel | C>T | p.P192S | VUS | 24.1 | Probably damaging | EUR | Het | 1.80E-06 |
| GRN | chr17:44350801 | Novel | G>T | c.708+1G>T | - | 32 | - | EUR | Het | 1.81E-06 |
| GRN | chr17:44351082 | Novel | ->TG | p.V252fs | - | 25.8 | - | EUR | Het | - |
| GRN | chr17:44351610 | Novel | A>G | p.K332E | VUS | 20.3 | Benign | EUR | Het | 3.60E-06 |
| GRN | chr17:44351663 | Novel | ->G | p.P349fs | - | 23 | - | EUR | Het | 8.99E-07 |
| GRN | chr17:44352025 | Novel | G>A | p.C397Y | VUS | 26.4 | Probably damaging | EUR | Het | 9.00E-07 |
| GRN | chr17:44352249 | Novel | G>A | c.1413+1G>A | - | 35 | - | EUR | Het | 9.01E-07 |
| GRN | chr17:44352395 | rs886053006 | G>A | p.V490M | VUS | 25.4 | Probably damaging | EUR | Het | 3.39E-06 |
| MAPT | chr17:45962447 | rs966689443 | G>C | p.G37A | VUS | 21.6 | Probably damaging | EUR | Het | 3.56E-05 |

**Table 2 (continued) | Discovery phase: Multi-ancestry summary of novel potential disease-causing variants only present in Alzheimer's disease and related dementia cases in AoU, 100KGP and UKB**

| Gene | Position | rs ID/ClinVar ID | cDNA changes | Protein change/ Splicing | Clinical significance | CADD | PP2 prediction | Genetic ancestry | Zygosity | GnomAD |
|---|---|---|---|---|---|---|---|---|---|---|
| MAPT | chr17:45978420 | rs139796158 | C>G | p.A60G | VUS | 25.1 | Probably damaging | AFR | Het | 5.64E-04 |
| MAPT | chr17:45978422 | Novel | G>T | p.A61S | VUS | 23.9 | Probably damaging | EUR | Het | 1.80E-06 |
| MAPT | chr17:45982886 | rs940936590 | C>T | p.R103W | - | 20.5 | - | EUR | Het | 1.93E-05 |
| MAPT | chr17:45983453 | rs2073193780 | G>A | p.E292K | VUS | 23.1 | Possibly damaging | EUR | Het | 8.50E-07 |
| MAPT | chr17:45983504 | Novel | C>- | p.P309fs | - | 22.5 | - | EUR | Het | - |
| MAPT | chr17:45996504 | rs779901466 | G>A | p.R163Q | VUS | 29.3 | Probably damaging | EUR | Het | 9.45E-06 |
| MAPT | chr17:45996630 | Novel | C>T | p.T205I | VUS | 27.5 | Probably damaging | EUR | Het | 8.99E-07 |
| MAPT | chr17:46010394 | Novel | G>A | p.G245S | VUS | 33 | Probably damaging | EUR | Het | 3.67E-06 |
| MAPT | chr17:46018621 | Novel | G>A | p.G245D | VUS | 33 | Probably damaging | EUR | Het | 9.03E-07 |
| MAPT | chr17:46018639 | Novel | A>G | p.K251R | VUS | 24.7 | Probably damaging | EUR | Het | - |
| MAPT | chr17:46024019 | rs991713081 | A>G | p.I303V | VUS | 26.2 | Probably damaging | EUR | Het | 4.50E-06 |
| APP | chr21:25891742 | Novel | T>G | p.I600L | VUS | 25.4 | - | EUR | Het | 5.40E-06 |
| APP | chr21:25905045 | rs768182065 | G>C | p.R517G | VUS | 25.8 | Probably damaging | SAS | Het | 3.48E-05 |
| APP | chr21:25905048 | rs368159818 | C>T | p.D516N | VUS | 27 | Probably damaging | EUR | Het | 5.09E-06 |
| APP | chr21:25911879 | rs201874897 | C>T | p.D460N | VUS | 26.9 | Probably damaging | EUR | Het | 5.09E-06 |
| APP | chr21:25911885 | rs755645885 | C>T | p.G458R | VUS | 27.8 | Probably damaging | EUR | Het | 1.19E-05 |
| APP | chr21:25954659 | rs200500889 | G>A | p.R409C | VUS | 32 | Probably damaging | EUR | Het | 6.30E-06 |
| APP | chr21:25982424 | rs752243493 | G>A | p.P251S | VUS | 26.4 | Probably damaging | EUR | Het | 1.10E-05 |
| APP | chr21:26000138 | rs200539466 | T>C | p.I248V | VUS | 23.1 | Probably damaging | EUR | Het | 1.35E-05 |
| APP | chr21:26021858 | rs772069024 | C>G | p.V227L | VUS | 25.4 | Probably damaging | EUR | Het | 8.99E-07 |
| APP | chr21:26021912 | rs754672142 | C>T | p.A209T | VUS | 20.9 | Probably damaging | EUR | Het | 4.50E-06 |
| APP | chr21:26051100 | rs199744129 | G>A | p.P132S | VUS | 27.9 | Probably damaging | EUR | Het | 1.70E-06 |
| SNCA | chr4:89726638 | rs746232417 | G>T | p.P90H | Likely pathogenic | 24.6 | Probably damaging | EUR | Het | 2.71E-06 |
| SNCA | chr4:89822281 | Novel | C>A | p.A91S | Likely pathogenic | 23.8 | Probably damaging | EUR | Het | 1.80E-06 |

**Table 2 (continued) | Discovery phase: Multi-ancestry summary of novel potential disease-causing variants only present in Alzheimer's disease and related dementia cases in AoU, 100KGP and UKB**

| Gene | Position | rs ID/ClinVar ID | cDNA changes | Protein change/Splicing | Clinical significance | CADD | PP2 prediction | Genetic ancestry | Zygosity | GnomAD |
|---|---|---|---|---|---|---|---|---|---|---|
| TREM2 | chr6:41161292 | Novel | A>C | p.L121R | VUS | 27 | Probably damaging | EUR | Het | 7.19E-06 |
| TREM2 | chr6:41161343 | Novel | T>A | p.D104V | VUS | 23.7 | Probably damaging | EUR | Het | 8.99E-07 |
| APOE | chr19:44907903 | rs1265280650 | G>C | p.E89Q | VUS | 22.3 | Possibly damaging | EUR | Het | 3.39E-06 |
| APOE | chr19:44909042 | rs765845034 | A>G | p.E275G | VUS | 22.5 | Probably damaging | EUR | Het | 8.66E-07 |
| APOE | chr19:44909029 | rs1334498236 | G>A | p.D271N | VUS | 22.8 | Probably damaging | SAS | Het | 1.25E-05 |
| APOE | chr19:44909171 | rs121918398 | G>A | p.R318H | Pathogenic/VUS | 23.6 | Probably damaging | EUR | Het | 6.30E-06 |
| APOE | chr19:44908777 | Novel | A>G | p.K187E | VUS | 24.8 | Probably damaging | EUR | Het | 9.20E-07 |
| APOE | chr19:44908625 | Novel | G>C | p.R136P | VUS | 25.8 | Probably damaging | EUR | Het | 9.02E-07 |
| APOE | chr19:44907945 | rs770545391 | G>A | p.E103K | VUS | 26.8 | Probably damaging | EUR | Het | 3.60E-06 |
| APOE | chr19:44907831 | Novel | C>T | p.Q65X | Likely pathogenic | 34 | - | AFR | Het | 2.99E-05 |
| TARDBP | chr1:11022581 | rs1643663101 | C>G | p.A391G | VUS | 20.5 | Benign | EUR | Het | - |
| TARDBP | chr1:11013936 | Novel | A>G | p.N70S | VUS | 21 | Benign | EUR | Het | 7.19E-06 |
| TARDBP | chr1:11022211 | Novel | A>G | p.R268G | VUS | 26.2 | Possibly damaging | EUR | Het | 8.99E-07 |
| TARDBP | chr1:11022617 | Novel | C>G | p.S403X | VUS | 37 | - | EUR | Het | - |
| TBK1 | chr12:64481955 | rs1381192966 | T>C | p.M309T | VUS | 20.2 | Benign | EUR | Het | 1.80E-06 |
| TBK1 | chr12:64490055 | Novel | T>C | p.M486T | VUS | 20.5 | Benign | EUR | Het | 9.01E-07 |
| TBK1 | chr12:64464391 | Novel | A>G | p.T96A | VUS | 23.1 | Benign | EUR | Het | 5.41E-06 |
| TBK1 | chr12:64467004 | Novel | A>C | p.K154N | VUS | 23.3 | Probably damaging | EUR | Het | 1.80E-06 |
| TBK1 | chr12:64497733 | Novel | C>T | p.T682I | VUS | 23.6 | Benign | EUR | Het | 2.71E-06 |
| TBK1 | chr12:64484368 | rs753694282 | T>C | p.I353T | VUS | 23.8 | Probably damaging | EUR | Het | 6.30E-06 |
| TBK1 | chr12:64484328 | Novel | T>C | p.Y340H | VUS | 24.1 | Probably damaging | EUR | Het | 3.60E-06 |
| TBK1 | chr12:64464434 | Novel | C>A | p.S110Y | VUS | 24.5 | Benign | EUR | Het | 9.08E-07 |
| TBK1 | chr12:64455884 | Novel | C>A | p.S5Y | VUS | 24.7 | Probably damaging | SAS | Het | 2.33E-05 |
| TBK1 | chr12:64481994 | rs751180634 | A>T | p.H322L | VUS | 25.2 | Possibly damaging | EUR | Het | 9.05E-06 |
| TBK1 | chr12:64495773 | Novel | G>C | p.R573P | VUS | 26 | Probably damaging | EUR | Het | 9.14E-07 |
| TBK1 | chr12:64495549 | rs2040916858 | G>A | p.G530R | VUS | 26.7 | Probably damaging | EUR | Het | 8.99E-07 |

**Table 2 (continued) | Discovery phase: Multi-ancestry summary of novel potential disease-causing variants only present in Alzheimer's disease and related dementia cases in AoU, 100KGP and UKB**

| Gene | Position | rs ID/ClinVar ID | cDNA changes | Protein change/Splicing | Clinical significance | CADD | PP2 prediction | Genetic ancestry | Zygosity | GnomAD |
|---|---|---|---|---|---|---|---|---|---|---|
| TBK1 | chr12:64464416 | Novel | C>T | p.A104V | VUS | 27.9 | Probably damaging | EUR | Het | 1.18E-05 |
| | chr12:64497966 | Novel | A>G | c.2067-2A>G | - | 29.3 | - | EUR | Het | 2.71E-06 |
| TBK1 | chr12:64481940 | rs904427940 | A>T | p.D304V | VUS | 29.8 | Probably damaging | EUR | Het | 5.94E-06 |
| | chr12:64464371 | rs2040580815 | G>A | p.C89Y | VUS | 31 | Possibly damaging | EUR | Het | - |
| TBK1 | chr12:64496367 | Novel | ->AAAATTCTTC | p.R574fs | - | 34 | - | EUR | Het | - |
| TBK1 | chr12:64496951 | Novel | A>- | p.Q588fs | - | 34 | - | EUR | Het | - |

The frequency of gnomAD refers to the frequency in the ancestry where the variants were found. Bold variants were replicated in different databases. Novel in the title means it has not been reported for the disease.

cDNA complementary DNA, VUS variant uncertain significance, Het Heterozygous, CADD Combined Annotation Dependent Depletion, PP2 PolyPhen-2, Clinical significance based on dbSNP, ClinVar, and ACMG guideline. Position refers to GRCh 38. AoU All of Us, 100KGP 100,000 Genomes Project, UKB UK Biobank, EUR European, AFR African, AMR American Admixed, AAC African Admixed, AJ Ashkenazi Jewish, EAS Eastern Asian, SAS South Asian, MDE Middle Eastern, FIN Finnish, CAH Complex Admixture History.

<sup>a</sup>These variants were not replicated only in cases across the diverse biobanks in the discovery phase. For AFR/AAC, frequencies in gnomAD for African/African American populations were used.

suggested to be AD, DLB, and argyrophilic grain disease[45]. We identified this variant in four European AD patients, three of whom presented with early onset in their fifties. Interestingly, we also identified this variant in a DLB patient in her early 60s. Neuropathological data and McKeith criteria[46] strongly supported a diagnosis of DLB in this patient. Although this variant has been widely reported across different types of dementia, our finding is the first report of this variant in DLB with a McKeith criteria of "high likelihood of DLB," expanding the etiological spectrum of *GRN* variation (Supplementary Data 7).

*GRN* p.C222Y was previously reported in a familial AD case from Latin American (Caribbean Hispanic) ancestry[47,48]. While the AAO for this patient was not reported, the mean AAO for the cohorts under study was 56.9 years (SD = 7.29), with a range of 40–73 years. In our study, we identified this variant in an individual of American Admixed ancestry with dementia in his late 40s and a disease duration of 11 years to date. This finding reinforces the role of this variant in early-onset disease.

There are several other interesting findings regarding variants in *GRN*. The *GRN* c.708+1G>A variant was previously reported in several FTD, FTLD, and corticobasal syndrome (CBS) cases, mostly early-onset[43,49]. We identified this variant in two European AD cases, both diagnosed in their 70s, marking the first report of this variant in late-onset Alzheimer's disease (LOAD). The *GRN* p.P166fsX variant was previously reported in an early-onset behavioral variant FTD case[50]. In our study, we identified this variant in a European dementia case diagnosed in her mid 70s with a disease duration of eight years to date. Furthermore, the *GRN* p.R418X variant is identified in the literature in two cases of FTLD with ubiquitin-positive inclusions (FTLD-U) with AAOs of 49 and 60 years[51]. We identified this variant in a European dementia case in her early 70s. Both findings represent the first report of these variants in late-onset dementia.

*PSEN1* R269H is a known pathogenic variant causing early-onset Alzheimer's disease (EOAD)[52,53]. However, it has been previously reported in only two LOAD cases[54,55]. In our study, we identified this variant in European and African Admixed ancestries in a total of 12 cases (eight AD and four related dementias), six of which were early-onset (≤65 years) and six were late-onset (>65 years). This finding underscores the potential for *PSEN1* p.R269H to contribute to LOAD with reduced penetrance. Additionally, one EOAD case that presented with hallucinations[56] and another case that manifested a behavioral presentation[57] have been reported to carry this variant. In this study, we identified *PSEN1* p.R269H in one FTD patient in the 100KGP cohort, marking the first report of this variant in FTD.

*MAPT* p.R406W has been reported in several familial cases of FTD with parkinsonism, all with early onset[58]. There are only two articles related to this variant in AD. The first describes a family with AD-like symptoms, with an average AAO of 61 years[59], and the other reports a familial AD case with an AAO of 50 years[60]. In our study, we identified this variant in nine AD cases, with a mean AAO of 61 years. This finding underscores the role of this variant in EOAD.

*TARDBP* p.G287S has been reported as a cause of ALS[23]. Here, for the first time, we identified this variant in two early-onset dementia cases in the AoU and 100KGP datasets.

*TBK1* p.D639H was previously reported in EOAD[61]. Here, we identified this variant in a LOAD patient in her late 70s. *TBK1* p.N22H, p.R228H, and p.E476fs were previously reported in ALS, ALS, and motor neuron disease, respectively[62,63]. We identified these variants in AD cases for the first time.

Several variants in *GBA1*, such as p.F298L, p.V230G, p.W223R, and p.P198L, have been previously reported in Gaucher disease patients. In our study, we identified these variants in heterozygous state in one AD case and five dementia cases, all with late onset. *GBA1* p.W223R was found in one AD case of Complex Admixture History ancestry. *GBA1* mutations are known to confer an increased risk for dementia in PD

and DLB. Notably, they have not been previously suggested to contribute to AD.

Similarly, *APP* p.E693Q has been reported in a few AD cases. In our study, we identified it in a related dementia case and no controls. This finding suggests that this variant may also be implicated in other types of dementia.

Several known variants identified in this study confirm previous findings related to disease type and onset. For example, the *APP* p.V717L variant has been reported in numerous AD cases, primarily in early-onset forms[64,65]. In our study, we identified this variant in two cases of EOAD with AAO ranges of 51–55 years and 56–60 years, respectively. Additional examples are reported in Supplementary Data 7 and 8.

In AoU, the *SNCA* p.Q99R variant was found in a female patient diagnosed in her late 60s with unspecified dementia without behavioral disturbance. In ADSP, the variant was identified in a male patient with pure AD in his early 70s. *SNCA* p.P90H and p.A91S were found in two males in their late 70s in the UKB cohort. All four patients were of European ancestry. Previously reported mutations in *SNCA* are known to cause early-onset PD and DLB[66,67]. The mean AAO in patients carrying *SNCA* mutations in this study is 72.75 years. These data suggest that these variants may not be disease-causing but could represent rare risk factors given their absence in controls and the replication of p.Q99R across datasets.

Novel variants found in this study that may potentially be associated with early-onset dementia include: p.L597W, p.V375I, p.L364F, p.A209T, p.D460N, p.R409C, and p.V227L variants in *APP;* p.R54X and p.M457V in *PSEN1;* p.H169R, p.D320N, and p.G349R in *PSEN2;* p.R556C and p.V28fs in *GRN;* p.G332LfsX64 and p.G701R in *MAPT;* p.G103D and

p.A42T in *GBA1;* p.R556C in *TARDBP,* p.N42S and p.R507I in *TBK1;* p.E275G and p.D271N in *APOE* and *TREM2* p.W44X. Among these variants, *APP* p.L364F and *PSEN1* p.R54X were found in vascular dementia cases with AAO ranges of 56–60 years and 41–45 years, respectively. Additionally, *PSEN2* p.D320N was found in an FTD case with an AAO of mid-50s. Another notable finding is the identification of *GRN* p.R556C in a dementia case with an AAO of mid-30s.

## *APOE* drives different population-attributable risk for Alzheimer's disease and related dementias

We separately analyzed ancestry-specific effects of *APOE* on AD/ADRDs. The summary of these findings is depicted in Fig. 4, Supplementary Data 9, and Supplementary Fig. 3.

In AoU, UKB, and 100KGP datasets, the *APOE* ε4/ε4 genotype exhibits a higher frequency among both AD patients and control individuals of African and African Admixed ancestries compared to Europeans. Related dementia patients show similar results in UKB. In AoU, related dementia cases of African Admixed ancestry show a higher frequency than cases of European ancestry, while case frequencies are similar between Africans and Europeans, likely due to the limited number of individuals with African ancestry in this dataset. In ADSP, the frequencies of this genotype among AD patients are similar across the three ancestries. Among control individuals in ADSP, *APOE* ε4/ε4 is more frequent in African Admixed and African ancestries than in Europeans, as previously reported[68]. Notably, the *APOE* ε4/ε4 genotype was absent from African and African Admixed DLB cases and controls in the AMP PD dataset. The frequency of *APOE* ε4/ε4 in Europeans was higher in cases compared to controls in AMP PD.

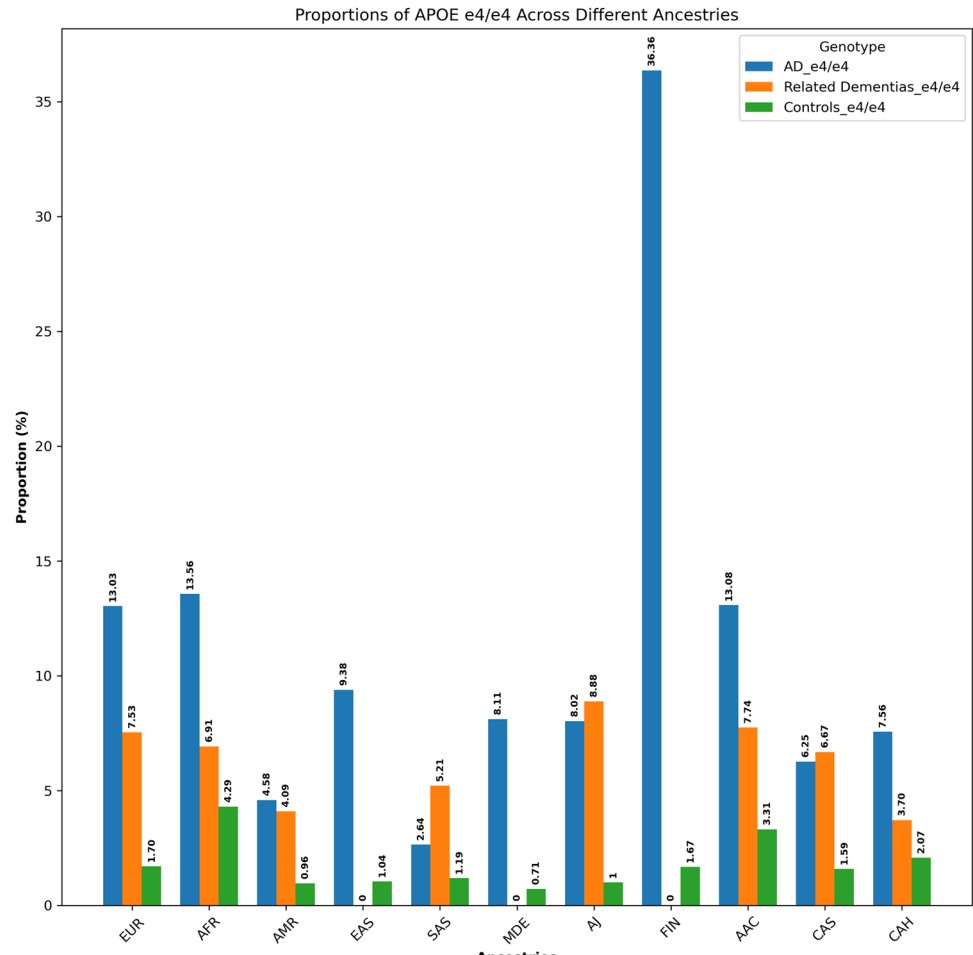

**Fig. 4 |** Proportions of *APOE* ε4/ε4 across 11 genetic ancestries in Alzheimer's disease, related dementias, and controls in all datasets.

When combining results across all datasets, the frequency of *APOE* ε4/ε4 in African and African Admixed AD patients was still higher than in Europeans, but the values were not significantly different. However, the frequency of *APOE* ε4/ε4 in control individuals of African and African Admixed ancestries was substantially higher than in controls of European ancestry. Additionally, the frequency of *APOE* ε4/ε4 in Finnish individuals was higher in AD cases and lower in controls compared to Europeans.

## Disease-modifying variants in *APOE* ε4 carriers modulate Alzheimer's and dementia risk across different ancestries

The summary of our findings for the frequencies of protective and disease-modifying variants under study, alongside *APOE* genotypes across all five datasets, is depicted in Supplementary Data 10–14. The proportions of individuals carrying *APOE* ε4 homozygous or heterozygous genotypes alongside protective or disease-modifying variants, within the total population, total ε4/ε4 carriers, and total ε4 carriers across each ancestry, combined across all biobanks, in AD, related dementias, and controls are reported in Supplementary Fig. 4 and Supplementary Data 15. Summaries of our findings for all the assessed models in *APOE* ε4, ε4ε4, and ε3ε3 are shown in Supplementary Fig. 5, and Supplementary Data 16–25.

In brief, we observe higher frequencies of individuals carrying *APOE*:rs449647-T, *19q13.31*:rs10423769-A, *APP*:rs466433-G, or *APP*:rs364048-C protective variant alleles alongside either one or two copies of *APOE* ε4 among African and African Admixed ancestries compared to Europeans for AD, related dementias, and controls. Carriers of *APOE*:rs449647-T and *19q13.31*:rs10423769-A are particularly noteworthy because *APOE*:rs449647-T displays the highest frequency among these ancestries, and the ratio of frequencies for *19q13.31*:rs10423769-A in both African and African Admixed ancestries compared to Europeans is substantially higher than the other protective/disease-modifying variants investigated across all three cohorts. In individuals of African ancestry, *19q13.31*:rs10423769-A was found to have a higher frequency in controls compared to both AD and related dementia cases among *APOE* ε4 homozygous or heterozygous carriers. In contrast, *APOE*:rs449647-T was found to have a lower frequency in controls compared to AD cases among *APOE* ε4 homozygous or heterozygous carriers in African ancestry but showed a higher frequency in controls carrying *APOE* ε4/ε4 versus cases in European populations (Supplementary Data 15).

The combination of *TOMM40*:rs11556505-T with either homozygous or heterozygous *APOE* ε4 was observed to have a higher frequency in Europeans and a lower frequency in Africans compared to most other ancestries across all three phenotypes. Additionally, the combination of *NOCT*:rs13116075-G and *APOE* ε4 homozygous or heterozygous was found to have a higher frequency in individuals of European and African Admixed ancestry versus Africans, in AD cases as compared to controls.

The protective model shows a modifying effect of *APOE*:rs449647-T in European, African, and Ashkenazi Jewish ancestries as well as a modifying effect of *TOMM40*:rs11556505-T in European, American Admixed, Ashkenazi Jewish, African Admixed, and individuals of Complex Admixture History. The R² model indicates that these variants are not in linkage disequilibrium with the *APOE* risk variants rs429358 and rs7412.

Significant interactions were found between *APOE* ε4 and the following variants: *19q13.31*:rs10423769-A in Africans; *NOCT*:rs13116075-G in both African and African Admixed populations; *CASS4*:rs6024870-A in the Complex Admixed History group; *LRRC37A*:rs2732703-G in the American Admixed ancestry; *APOE*:rs449647-T in European and African Admixed ancestries; and *TOMM40*:rs11556505-T in Europeans. In *APOE* ε4/ε4 carriers, the interaction with *APOE*:rs449647-T was found to be significant in American Admixed and African Admixed populations, while *TOMM40*:rs11556505-T was significant in the European ancestry.

The interaction model in *APOE* ε3/ε3 shows no significant *p* value for *19q13.31*:rs10423769-A in Africans, *NOCT*:rs13116075-G in African Admixed ancestry, *CASS4*:rs6024870-A in Complex Admixed History, and *LRRC37A*:rs2732703-G in American Admixed ancestry, but highly significant *p* values for *APOE*:rs449647-T and *TOMM40*:rs11556505-T in European and *NOCT*:rs13116075-G in Africans, with opposite directional effects compared to *APOE* ε4 carriers. These data confirm the role of these variants in modulating the effect of *APOE* ε4 in AD risk.

To assess the potential enrichment of protective or disease-modifying variants in cases versus controls, we conducted logistic regression on individuals with the highest genetic risk burden based on the PRS, including 1,997 cases and 559 controls. Results are presented in Supplementary Data 26. The analysis revealed no significant differences between cases and controls for these variants and indicated no interaction between these variants and PRS to modify disease penetrance. This analysis may be underpowered due to the low frequency of these variants.

We performed a burden analysis to determine whether there was an enrichment of predicted pathogenic variants in cases compared to controls by gene, after excluding known variants. Burden testing, adjusted for sex, age, and principal components (PCs 1–10) using SKAT-O, revealed an enrichment of rare variants in *GBA1* ($P = 0.023$) (Supplementary Data 27 and 28).

## Discussion

We undertook a large and comprehensive characterization of potential disease-causing, risk, protective, and disease-modifying variants in known AD/ADRDs genes, aiming to create an accessible genetic catalog of both known and novel coding and splicing variants associated with AD/ADRDs in a global context. Our results expand our understanding of the genetic basis of these conditions, potentially leading to new insights into their pathogenesis, risk, and progression. A comprehensive genetic catalog can inform the development of targeted therapies and personalized medicine approaches in the new era of precision therapeutics.

We present a user-friendly platform for the AD/ADRDs research community that enables easy and interactive access to these results (https://niacard.shinyapps.io/MAMBARD_browser/). This browser may serve as a valuable resource for researchers, clinicians, and clinical trial designers.

We identified 156 genetic variants (23 known and 133 novel) in AD/ADRDs across diverse populations. The successful replication of novel variants across different datasets increases the likelihood of these variants being pathogenic and warrants further validation through future functional studies, highlighting their potential broader applicability and significance in global genetics research.

We identified 26 potentially disease-causing variants in non-European ancestries, including 18 that were absent in individuals of European ancestry. This highlights the necessity of expanding genetics research to diverse populations, corroborating the notion that the genetic architecture of AD/ADRDs risk differs across populations. We identified a total of 23 variants in control individuals that had been previously reported as disease-causing. This scenario involves three possibilities: (i) the mutation is a non-disease-causing variant found by chance in an ADRD patient, (ii) the mutation is pathogenic but exhibits incomplete penetrance, or (iii) control individuals represent undiagnosed patients. These findings reveal the potential for conflicting reports and misinterpretations, emphasizing the need for careful analysis and functional validation in genetics research. It underscores the critical importance of caution in identifying and interpreting potentially-pathogenic variants, which is essential for ensuring accurate diagnosis, risk assessment, genetic counseling, and development of effective treatments.

We explored genotype-phenotype correlations for both known and novel variants. Our findings involving known variants largely

reinforce previous studies, while expanding the clinical spectrum for various types of dementia, exhibiting different AAOs and/or additional clinical features not previously reported. While the genotype-phenotype correlations for newly identified variants require further investigation to fully understand their impact, we identified 24 novel variants that may potentially be associated with early-onset dementia and therefore warrant further study.

While several studies have conducted *APOE* genotyping across different age groups, sexes, and population ancestries[69,70], the differential role of *APOE* across 11 ancestries has not yet been explored in a global context. We found that African and African Admixed ancestry populations harbor a higher frequency of *APOE* ε4/ε4 carriers than populations of European ancestry. Recent studies have shed light on the varying risk associated with *APOE* ε4 alleles in populations of African ancestry. Indeed, it has been reported that individuals of African descent who carry the *APOE* ε4 allele have a lower risk of developing AD compared to other populations with the same allele. This suggests that the genetic background of African ancestry around the *APOE* locus is linked to a reduced odds ratio for risk variants[68]. Furthermore, a recent study has suggested the presence of a resilient locus (*19q13.31*) potentially modifying *APOE* ε4 risk in African-descent populations. This disease-modifying locus, located 2MB from *APOE*, significantly lowers the AD risk for African *APOE* ε4 carriers, reducing the magnitude of the effect from 7.2 to 2.1[10]. Our finding is in concordance with a large AD meta-analysis conducted to date[68]. We identified several variants with high frequency among *APOE* ε4 homozygous or heterozygous carriers in African and African Admixed ancestries. Notably, individuals carrying both *APOE* ε4 homozygous or heterozygous and either *APOE*:rs449647-T or *19q13.31*:rs10423769-A exhibit higher frequencies in African and African Admixed ancestries compared to Europeans.

Considering all the models under study, we find that in the presence of *APOE* ε4, *APOE*:rs449647-T decreases the risk of AD in Europeans but increases it in Africans. *TOMM40*:rs11556505-T increases the risk of AD in Europeans. *TOMM40*:rs11556505-T also increases the risk of AD in American Admixed and Ashkenazi Jewish ancestries, though not through an interaction with *APOE*. An interaction effect with *APOE* was found for *19q13.31*:rs10423769-A, *NOCT*:rs13116075-G, *CASS4*:rs6024870-A, and *LRRC37A*:rs2732703-G. Specifically, *19q13.31*:rs10423769-A reduces the risk of AD in Africans, *NOCT*:rs13116075-G reduces the risk in Africans but increases it in African Admixed ancestry, *CASS4*:rs6024870-A reduces the risk in Complex Admixed History ancestry, and *LRRC37A*:rs2732703-G increases the risk in American Admixed ancestry.

Our findings support previous literature, which indicates that *19q13.31* is an African-ancestry-specific locus that reduces the risk effect of *APOE* ε4 for developing AD. *APOE*:rs449647-T is a polymorphism in the regulatory region of *APOE* that can modulate the risk of developing AD by altering its affinity to transcription factors, thus affecting gene expression. Our study demonstrates rs449647-T's association with an increased risk of AD in *APOE* ε4 carriers of African ancestry and a decreased risk in *APOE* ε4 carriers of European ancestry. *TOMM40*:rs11556505-T has been variably associated with both risk and protective effects, likely depending on the phenotype being evaluated[11]. We show an association with an increased risk of AD in *APOE* ε4 carriers, particularly in Europeans. Of note, *APOE* and *TOMM40* are located in close proximity on chromosome 19q13.32, with *TOMM40* positioned immediately upstream of *APOE*. In addition, this study reveals the disease-modifying effect of *NOCT*:rs13116075-G, *CASS4*:rs6024870-A and *LRRC37A*:rs2732703-G across different ancestries. The interaction of these variants with *APOE* ε4 is not known, but identifying the mechanism(s) conferring protection could provide greater insights into the etiology of AD and inform potential ancestry-specific therapeutic interventions.

This comprehensive genetic characterization for AD/ADRDs across diverse populations holds critical implications for potential clinical trials and therapeutic interventions in a global context, highlighting its significance for such efforts worldwide. For example, clinical trials for *GRN* have recently commenced[71] (https://www.theaftd.org/posts/1ftd-in-the-news/b-ftd-grn-gene-therapy-abio/). Understanding population-specific frequencies of genetic contributors to disease is vital in the design and implementation of clinical trials for several reasons. Firstly, it allows for targeted recruitment, ensuring that studies include an adequate number of participants concordant with their genetic makeup. Secondly, it promotes diversity in clinical trial populations, which is essential to understand disease globally, as well as treatment responses across different groups. Furthermore, knowledge of population-specific variant frequencies may inform treatment efficacy assessments in the future, as genetics may influence treatment outcomes. In summary, it plays a vital role in personalized medicine, guiding more targeted and effective treatments based on individuals' genetic profiles.

Despite our efforts, there are several limitations and shortcomings to consider in this study. A major limitation is the under-representation of certain populations and the lack of clinical follow-up on cases, both of which lead to underpowered datasets and limit the possibility of drawing firm conclusions. It is our hope that the interactive tool we have developed will help the research community further investigate these mutations in clinical and longitudinally characterized cohorts in a global context. Additionally, the reliance on currently available datasets may introduce biases, as these datasets often have varying levels of coverage, quality, and accurate clinical information. One critical concern is the potential for misdiagnosis across different populations, which can result from overlapping clinical presentations and the inclusion of individuals with mixed or alternative dementia etiologies. This issue highlights the challenges in delineating clear diagnostic borders in dementia research, underscoring the need for neuropathologically confirmed cohorts and reinforcing the importance of blended diagnosis in future genetic analyses[72]. Additionally, inter-ancestry differences in diagnostic criteria exist. For instance, optimal cutoffs for the Montreal Cognitive Assessment require stratification by race, ethnicity, and education[73]. Other relevant considerations include the differing exclusion criteria for controls across cohorts and the limitations of genetically predicted subpopulations, which may affect the comparability of results. Future research should aim to include more diverse populations and improve the quality of genetic data in addition to standardized data harmonization efforts. Moreover, functional validation of identified variants is necessary to confirm their pathogenicity and relevance to AD/ADRDs. In addition, the current study did not focus on the *C9ORF72* repeat expansion due to its complex nature, as current approaches using short-read whole genome sequencing (WGS) data do not accurately detect it. Future studies should address this limitation.

Lastly, ongoing collaborations between researchers, clinicians, and policy-makers are crucial to ensure that advancements in genetic research translate into equitable and effective clinical applications. Our study is a step towards addressing these limitations by providing a more diverse genetic characterization and highlighting the need for inclusive research practices. Future research should continue to focus on enhancing the robustness and applicability of genetic findings in AD/ADRDs research, making knowledge globally relevant.

## Methods
### Ethical approval
Research conducted on biobank-scale data was deemed "not human subjects research" by the NIH Office of IRB Operations, and it was stated that no IRB approval is required. The NIH Intramural IRB also waived ethical approval for the overall study.

Demographic information, including age and sex, was provided in the self-reported survey in AoU and through the UKB, ADSP, and AMP PD portals. Self-reported demographic data of participants in 100KGP were obtained from Data Release V18 (12/21/2023) using the LabKey application incorporated into the research environment.

### Discovery phase: All of Us

The AoU Research Program (allofus.nih.gov) launched by the United States National Institutes of Health (NIH) endeavors to enhance precision health strategies by assembling rich longitudinal data from over one million diverse participants in the United States. The program emphasizes health equity by engaging underrepresented groups in biomedical research. This biobank includes a wide range of health information, including genetic, lifestyle data, and electronic health records (EHRs), among others, making it a valuable resource for studying the genetic and environmental determinants of various diseases including AD/ADRDs[6,74].

**Data acquisition.** We accessed the AoU data (v7) through the AoU researcher workbench cloud computing environment (https://workbench.researchallofus.org/), utilizing Python and R programming languages for querying. We used the online AoU data browser (https://databrowser.researchallofus.org/variants) to extract genetic variants from short-read WGS data. The selected variants were filtered for protein-altering or splicing mechanisms for further analysis.

**Cohort creation.** We generated WGS cohorts using the cohort-creating tool in the AoU Researcher Workbench. AD/ADRD cases were selected based on the condition domain in the EHRs. Controls were selected among individuals ≥65 years old without any neurological condition in their EHRs, family history, or neurological history in their self-reported surveys. In total, 539 AD cases, 1655 related dementias, and 13,835 controls were included in the study.

**Whole genome sequencing protocol and quality control assessment.** WGS was conducted by the Genome Centers funded by the AoU Research Program[6,75], all of which followed the same protocols. Sequencing details are described elsewhere[76]. Phenotypic data, ancestry features, and PCs were generated using Hail within the AoU Researcher Workbench (https://support.researchallofus.org/hc/en-us/articles/4614687617556-How-the-All-of-Us-Genomic-data-are-organized). Ancestry annotation and relatedness were determined using the PC-relate method in Hail; Duplicated samples and one of each related participant pair with KINSHIP less than 0.1 being excluded from the Hail data[77] (https://support.researchallofus.org/hc/en-us/articles/4614687617556-How-the-All-of-Us-Genomic-data-are-organized). Flagged individuals and low-quality variants (qc.call_rate <0.90) were removed from the analysis.

**Variant filtering and analysis.** We utilized protein-altering and splicing variants ("WGS_EXOME_SPLIT_HAIL_PATH") for our analysis, obtained following the tutorial "How to Work with AoU Genomic Data (Hail–Plink) (v7)." The largest intervals for genomic positions were obtained from the UCSC Genome Browser (https://genome.ucsc.edu/).

Variant datasets were obtained as described in the related workspace (see "How to Work with AoU Genomic Data (Hail–Plink) (v7)" for further details). Genomic positions (GRCh38) for each gene were extracted from the Hail variant dataset. Variant-level quality assessments were applied as described in the Manipulate Hail Variant Dataset tutorial (see the "How to Work with AoU Genomic Data" workspace for further details). VCF files containing the cohorts in the current study were generated using BCFtools v1.12[78]. Allele frequency and zygosity of each resulting variant were calculated per ancestry using PLINK v2.0[79] in each of the AD, related dementias, and control cohorts.

### Discovery phase: UK Biobank

The UKB (https://www.ukbiobank.ac.uk/) is a large-scale biomedical dataset containing detailed genetic, clinical, and lifestyle information from over 500,000 participants aged 40–69 years in the United Kingdom. Each participant's profile includes a diverse array of phenotypic and health-related information. Additionally, the participants' health has been followed long-term, primarily through linkage to a wide range of health-related records, enabling the validation and characterization of health-related outcomes[7]. This dataset has been instrumental in advancing research on various health conditions, including AD/ADRDs, by facilitating large-scale genome-wide association studies and rare, deleterious variant analyses[7].

**Cohort creation.** We accessed UKB data (v18.1) (https://www.ukbiobank.ac.uk/) through the DNAnexus cloud computing environment, utilizing the Python programming language for querying. Three experimental cohorts were defined: AD, related dementias, and controls. The AD cohort was defined by the UKB field ID 42020, using diagnoses according to the UKB's algorithmically defined outcomes v2.0 (https://biobank.ndph.ox.ac.uk/ukb/refer.cgi?id=460). The related dementia cohort was defined by the UKB field ID 42018, using the UKB's algorithmically defined "Dementia" classification, with the added step of excluding any individuals in the aforementioned AD cohort. The control cohort includes individuals ≥65 years without any neurological condition or family history of neurological disorders. Relatedness was calculated with KING[80], and individuals closer than cousins were removed by KINSHIP > 0.0884 to ensure no pair of participants across all three cohorts were related. In total, 4225 AD cases, 5306 related dementias, and 56,741 controls were included in the study.

**Whole genome sequencing protocol and quality control assessment.** Sequencing was conducted using the NovaSeq 6000 platform[81]. These data were then analyzed with the DRAGEN v3.7.8 (Illumina, San Diego, CA, USA) software. Alignment was performed against the GRCh38 reference genome. Further details on quality control metrics can be found at https://biobank.ndph.ox.ac.uk/showcase/label.cgi?id=187.

**Data acquisition.** WGS data are stored in the UKB as multi-sample aggregated pVCF files, each representing distinct 20 kbp segments for all participants. Genomic ranges were defined for each gene of interest using Ensembl (https://useast.ensembl.org/index.html). Those pVCF files containing any variants within these genomic ranges were included for analysis. Left alignment and normalization were performed on each of these variants using BCFtools v1.15.1[82]. Then, ANNOVAR[83] was used to annotate the normalized variants.

**Variant filtering and analysis.** We filtered variants to include only those within our genes of interest, annotated as either protein-altering or splicing variants and present in any AD and/or related dementias cases. Allele frequency and zygosity of each resulting variant were calculated per ancestry using PLINK v2.0[79] in each of the AD, related dementias, and control cohorts.

### Discovery phase: 100,000 Genomes Project

The 100KGP (https://www.genomicsengland.co.uk/)[84] has sequenced and analyzed genomes from over 75,000 participants with rare diseases and family members. Early onset dementia (encompassing FTD) is one of the rare diseases studied by the Neurology and Neurodevelopmental Disorders group within the rare disease domain. Participants were recruited by healthcare professionals and researchers from 13 Genomic Medicine Centres in England. The probands were enrolled in the project if they or their guardian provided written consent for their samples and data to be used in research. Probands and, if feasible,

other family members were enrolled according to eligibility criteria set for certain rare disease conditions.

WGS data (v18) were utilized from 180 unrelated cases with early-onset dementia (encompassing FTD and prion disease) or PD with dementia phenotype and 3479 unrelated controls ≥65 years at the time of the analysis. Sequencing and quality control analyses for the 100KGP were previously described elsewhere[85] (https://re-docs.genomicsengland.co.uk/sample_qc/). Protein-altering or splicing variants were obtained using the Exomiser variant prioritization application[86]. Candidate variants were extracted from a multi-sample aggregated VCF provided in the Genomics England (GE) research environment.

### Replication phase: Alzheimer's Disease Sequencing Project

The ADSP (https://adsp.niagads.org/), supported by the National Institute on Aging and the National Human Genome Research Institute, aims to generate data associated with AD/ADRDs. This dataset includes genetic data from thousands of individuals with and without AD, facilitating the discovery of novel genetic risk factors and pathways underlying the disease.

We used data from the ADSP dataset (v4) for this study, data which included a total of 10,566 AD cases and 16,217 controls. The control cohort includes individuals ≥65 years without any neurological condition or family history of neurological disorders. Samples were excluded from further analysis if the sample call rate was less than 95%, the genetically determined sex did not match the sex reported in clinical data, or excess heterozygosity was detected (|F| statistics > 0.25). For quality control purposes, a minor allele frequency (MAF) threshold of 0.1% was used. The missingness rate and allele frequency of these variants were calculated for each ancestry using PLINK v2.0[79] and PLINK v1.9[87]. Variant quality control included removing variants with Hardy-Weinberg Equilibrium $P < 1 \times 10^{-4}$ in control samples, differential missingness by case-control status at $P \leq 1 \times 10^{-4}$, and non-random missingness by haplotype at $P \leq 1 \times 10^{-4}$. Relatedness was calculated with KING[80], and individuals closer than cousins were removed by KINSHIP > 0.0884. ADSP includes a range of cohorts, including extensive family cohorts and cohorts with PSP, corticobasal degeneration, MCI, and DLB patients. Only samples labeled as definite AD or control were included in this analysis. We meticulously screened for identified genetic variants with a CADD score > 20 that were present across any of the three discovery datasets (AoU, 100KGP, and UKB) in the ADSP cohort.

### Replication phase: Accelerating Medicines Partnership in Parkinson's disease

Accelerating Medicines Partnership (AMP) (https://fnih.org/our-programs/accelerating-medicines-partnership-amp/) is a public-private initiative that aims to transform the current model for developing new diagnostics and treatments by jointly identifying and validating promising biological targets for therapeutics. It was launched in 2014 by the NIH, the U.S. Food and Drug Administration, multiple biopharmaceutical and life science companies, and several non-profit organizations. AMP PD focuses on advancing research into PD-related disorders and leverages cutting-edge technologies and large-scale data analysis to identify key genetic variants, biomarkers, and therapeutic targets associated with PD-related disorders, with the ultimate goal of developing novel treatments and improving patient outcomes.

We used AMP PD Release 3 genomic data, focusing specifically on DLB cases and controls. Samples were excluded from further analysis if the sample call rate was less than 95%, the genetically determined sex did not match the sex reported in clinical data, or excess heterozygosity was detected (|F| statistics > 0.25). Variant quality control included removing variants with missingness above 0.05%. Relatedness was calculated with KING[80], and individuals closer than first cousins were removed by KINSHIP > 0.0884. After quality control and ancestry prediction, this dataset contains a total of 2530 DLB cases and

3270 controls, characterized as individuals ≥65 years without any neurological condition or family history of any neurological disorders. We screened for identified variants with a CADD score >20 that were present across all three discovery datasets (AoU, 100KGP, and UKB) within AMP PD. Allele frequency of these variants per ancestry was calculated using PLINK v2.0[79] and PLINK v1.9[87].

### Diagnostic criteria across biobanks

AoU: Diagnoses are primarily derived from EHRs and linked to ICD-9/ICD-10 codes.

UKB: Diagnoses are obtained through linkage to national health registries, including hospital episode statistics and death certificates. UKB's algorithmically-defined outcomes are the primary source for identifying AD/ADRDs cases.

100KGP: This project focuses on genomic data, with clinical diagnoses often drawn from medical records provided by referring clinicians. Diagnoses originate from hospitals in collaboration with GE, where expert neurologists evaluate patients and submit diagnostic data to GE.

ADSP: Diagnoses are often based on standardized research protocols such as NINCDS-ADRDA or DSM-IV/DSM-5 criteria. Many cases include neuropathological confirmation or biomarker evidence (e.g., CSF amyloid/tau levels).

AMPPD: While primarily focused on PD, cases of mixed dementia are identified through clinical evaluations and medical records. Cases overlapping with AD/ADRDs are typically based on clinical diagnoses supported by neurological assessments.

### Ancestry prediction analysis

All samples in AoU, UKB, ADSP, and AMP PD datasets underwent a custom ancestry prediction pipeline included in the GenoTools package (https://github.com/dvitale199/GenoTools)[88]. In brief, ancestry was defined using reference panels from the 1000 Genomes Project, the Human Genome Diversity Project, and an Ashkenazi Jewish population dataset. We used a panel of 4008 samples from 1000 Genomes Project and the Gene Expression Omnibus database (www.ncbi.nlm.nih.gov/geo; accession no. GSE23636) to define ancestry reference populations. The reference panel was then reduced to exclude palindromic SNPs (AT or TA or GC or CG). SNPs with MAF < 0.05, genotyping call rate <0.99, and HWE $P < 1E-4$ in the reference panel were further excluded. Variants overlapping between the reference panel SNP set and the samples of interest were then extracted. Any missing genotypes were imputed using the mean of that particular variant in the reference panel. The reference panel samples were split into an 80/20 train/test set, and then PCs were fitted using the set of overlapping SNPs described previously. The PCs were then transformed via UMAP to represent global genetic population substructure and stochastic variation. A classifier was then trained on these UMAP transformations of the PCs (linear support vector). Based on the test data from the reference panel and at 5-fold cross-validation, 11 ancestries were predicted consistently with balanced accuracies greater than 0.95.

Genetic ancestry in 100KGP was estimated by generating PCs for 1000 Genomes Project phase 3 samples and projecting all participants onto the super populations in the 1000 Genomes Project, as described elsewhere (https://re-docs.genomicsengland.co.uk/ancestry_inference/). Despite our efforts to utilize GenoTools, we encountered significant challenges during its implementation in GE's High-Performance Computing Cluster (HPC). Consequently, GenoTools and GE's HPC were incompatible in this context. PCA plots across all biobanks are shown in Supplementary Fig. 6.

### Evaluation of potential disease-causing mutations, risk factors, and disease risk modifiers across ancestries

In the discovery phase, variants were filtered out based on their presence in control individuals across biobanks. To prioritize potential

disease-causing mutations, we followed the American College of Medical Genetics and Genomics (ACMG) guidelines (https://wintervar.wglab.org/), leveraging existing clinical and population databases and pathogenicity predictors including the Human Gene Mutation Database (HGMD) (https://www.hgmd.cf.ac.uk/ac/index.php), dbSNP (https://www.ncbi.nlm.nih.gov/snp/), gnomAD (https://gnomad.broadinstitute.org/), ClinVar (https://www.ncbi.nlm.nih.gov/clinvar/), PolyPhen-2 (http://genetics.bwh.harvard.edu/pph2/), and CADD scores (GRCh38-v1.7) (https://cadd.gs.washington.edu/).

Secondly, we investigated *APOE*, the major risk factor for AD/ADRDs, across diverse ancestries. We used PLINK (v1.9 and v2.0)[79,87] to extract genotypes for two *APOE* variants, rs429358 (chr19:44908684-44908685) and rs7412 (chr19:44908821-44908823), as a proxy for *APOE* allele status (ε1, ε2, ε3, and ε4) in the AoU, UKB, ADSP, and AMP PD datasets. Data analysis was conducted as reported elsewhere (https://github.com/neurogenetics/APOE_genotypes). In the 100KGP dataset, *APOE* genotypes were analyzed using PLINK v2.0[79] in a multi-sample aggregated VCF provided in the GE research environment. Subsequently, we calculated the number of individuals with each genotype per ancestry and their frequency percentages.

Finally, we assessed disease modifiers for *APOE* ε4 homozygous and heterozygous carriers specifically. A total of 21 variants, previously identified as either protective ($n = 11$) or resilient ($n = 10$), were extracted from all datasets using the same protocol previously described. Among them, *ABCA7*:rs72973581-A, *APP*:rs466433-G, *APP*:rs364048-C, *NOCT*:rs13116075-G, *SORL1*:rs11218343-C, *SLC24A4*:rs12881735-C, *CASS4*:rs6024870-A, *EPHA1*:rs11762262-A, *SPPL2A*:rs59685680-G, *APP*:rs63750847-T, *PLCG2*:rs72824905-G are protective, while *19q13.31*:rs10423769-A, *APOE*:rs449647-T, *FN1*:rs140926439-T, *FN1*:rs116558455-A, *RELN*:rs201731543-C, *TOMM40*:rs11556505-T, *RAB10*:rs142787485-G, *LRRC37A*:rs2732703-G, *NFIC*:rs9749589-A, and the *APOE3* Christchurch:rs121918393-A variant are reported to be resilient. These variants were then checked across all *APOE* genotypes and ancestries. Carrier frequencies (either heterozygous or homozygous) were calculated for each *APOE* genotype and ancestry, and were then combined across each of the datasets. In AoU, a variant dataset in Hail format (WGS_VDS_PATH) was used for the analysis. PLINK v2.0[79] and R v4.3.1 (https://www.r-project.org/) were used to assess the protective model (which evaluates the effect of each protective/disease-modifying variant on the phenotype), conditional model (which evaluates the effect of each protective/disease-modifying variant on the phenotype in the presence of *APOE* (ε4, ε4/ε4, ε3/ε3)), $R^2$ model (which evaluates the correlation of each protective/disease-modifying variant with *APOE* (ε4, ε4/ε4, ε3/ε3)), and interaction model (which evaluates putative interactions between each protective/disease-modifying variant and *APOE* (ε4, ε4/ε4, ε3/ε3) on the phenotype). Logistic and linear regression analyses, adjusting for *APOE* status, sex, age, and PCs, were applied in the most well-powered dataset (ADSP) to explore these effects.

PRS was calculated using summary statistics from Kunkle 2019[89] for European ancestry samples in ADSP data. Individuals with the highest genetic risk burden (top 25th percentile), including both cases and controls, were then subsetted. A logistic regression analysis of all 21 protective/disease-modifying variants was performed using two separate models: one adjusted for Z score, sex, age, and PCs (1–5), and the other adjusted for Z score, *APOE* status, sex, age, and PCs (1–5), applied to the selected individuals.

A burden analysis for rare variants (MAF ≤ 0.01), adjusted for covariates, was performed using SKAT-O in Rvtests v2.1.0 on the 11 studied genes in the European ancestry group of the ADSP dataset. This analysis was conducted by functional category: the first analysis included only synonymous variants, while the second analysis excluded synonymous variants, as well as known coding and splicing variants listed in Table 1.

## Reporting summary

Further information on research design is available in the Nature Portfolio Reporting Summary linked to this article.

## Data availability

All data supporting the findings of this study are available within the paper and its Supplementary Information files.

## Code availability

The code used in this study can be found online at https://github.com/NIH-CARD/ADRD-GeneticDiversity-Biobanks, https://doi.org/10.5281/zenodo.13363465.

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

## Acknowledgements

We thank Paige Brown Jarreau for her meticulous editing of this manuscript. This research was supported [in part] by the Intramural Research Program of the National Institutes of Health (NIH); project number ZO1 AG000535 and ZIA AG000949. The contributions of the NIH author(s) were made as part of their official duties as NIH federal employees, are in compliance with agency policy requirements, and are considered Works of the United States Government. However, the findings and conclusions presented in this paper are those of the author(s) and do not necessarily reflect the views of the NIH or the U.S. Department of Health and Human Services. This work utilized the computational resources of the NIH HPC Biowulf cluster. (http://hpc.nih.gov). We have received an exception to the Data and Statistics Dissemination Policy from the All of Us Resource Access Board. The All of Us Research Program is supported by the National Institutes of Health, Office of the Director: Regional Medical Centers: 1 OT2 OD026549; 1 OT2 OD026554; 1 OT2 OD026557; 1 OT2 OD026556; 1 OT2 OD026550; 1 OT2 OD 026552; 1 OT2 OD026553; 1 OT2 OD026548; 1 OT2 OD026551; 1 OT2 OD026555; IAA #: AOD 16037; Federally Qualified Health Centers: HHSN 263201600085U; Data and Research Center: 5 U2C OD023196; Biobank: 1 U24 OD023121; The Participant Center: U24 OD023176; Participant Technology Systems Center: 1 U24 OD023163; Communications and Engagement: 3 OT2 OD023205; 3 OT2 OD023206; and Community Partners: 1 OT2 OD025277; 3 OT2 OD025315; 1 OT2 OD025337; 1 OT2 OD025276. In addition, the All of Us Research Program would not be possible without the partnership of its participants. This research has been conducted using the UK Biobank Resource under application number 33601. This research was made possible through access to data in the National Genomic Research Library, which is managed by Genomics England Limited (a wholly owned company of the Department of Health and Social Care). The National Genomic Research Library holds data provided by patients and collected by the NHS as part of their care and data collected as part of their participation in research. The National Genomic Research Library is funded by the National Institute for Health Research and NHS England. The Wellcome Trust, Cancer Research UK and the Medical Research Council have also funded research infra-structure. This research has been conducted using the Alzheimer's Disease Sequencing Project (ADSP) Resource under accession number NG00067. The data for this study were prepared, archived, and distributed by the National Institute on Aging Alzheimer's Disease Data Storage Site (NIAGADS) at the University of Pennsylvania (U24-AG041689), funded by the National Institute on Aging. This work was supported in part by the Intramural Research Program of the National Institute on Aging and the National Institute of Neurological Disorders and Stroke (project number Z01-AG000949-02). Data used in the preparation of this article were obtained from the AMP PD Knowledge

Platform. For up-to-date information on the study, please visit https://www.amp-pd.org. AMP PD—a public-private partnership—is managed by the FNIH and funded by Celgene, GSK, the Michael J. Fox Foundation for Parkinson's Research, the National Institute of Neurological Disorders and Stroke, Pfizer, Sanofi, and Verily. Clinical data and biosamples used in the preparation of this article were obtained from the Parkinson's Progression Markers Initiative (PPMI), and the Parkinson's Disease Biomarkers Program (PDBP). PPMI—a public-private partnership—is funded by the Michael J. Fox Foundation for Parkinson's Research and funding partners, including full names of all of the PPMI funding partners found at http://www.ppmi-info.org/fundingpartners. The PPMI Investigators have not participated in reviewing the data analysis or content of the manuscript. For up-to-date information on the study, visit http://www.ppmi-info.org. The Parkinson's Disease Biomarker Program (PDBP) consortium is supported by the National Institute of Neurological Disorders and Stroke (NINDS) at the National Institutes of Health. A full list of PDBP investigators can be found at https://pdbp.ninds.nih.gov/policy. The PDBP Investigators have not participated in reviewing the data analysis or content of the manuscript. PDBP sample and clinical data collection is supported under grants by NINDS: U01NS082134, U01NS082157, U01NS082151, U01NS082137, U01NS082148, and U01NS082133.

## Author contributions

S.B.C. contributed to the study concept or design. M.K., F.A., S.M.G., S.C.A., P.S.L., F.F., H.L., J.J.K., M.J.K., and M.B.M. were involved in the analysis of data across different biobanks. All authors contributed to the critical review and had final responsibility for the decision to submit for publication.

## Competing interests

F.F., H.L., M.J.K., M.B.M., and M.A.N.'s participation in this project was part of a competitive contract awarded to Data Tecnica LLC by the US National Institutes of Health (NIH).
