## [Transparent Peer Review file · Nature Communications]

Biobank-scale genetic characterization of Alzheimer's disease and related dementias across diverse ancestries

Corresponding Author: Dr Sara Bandres-Ciga

Version 0:

Reviewer comments:

Reviewer #1

(Remarks to the Author)

This ms represents an enormous amount of data collation and analysis which will be of value to the field (although the complexity of the analysis means that, necessarily I suspect much of the data will be difficult for researchers to find).

I have one major concern which relates to the quality of the diagnoses of "Alzheimer's disease" in general and across populations (in white US/UK populations the diagnosis is about 70% accurate... we have no idea how accurate it is in other populations). This is a serious concern which should be discussed. I note, for example, the occurrence of GRN mutations... which is an FTD gene. The recent paper on neuropath confirmed dementia cases could be cross referenced in a discussion of this issue (see bottom of review)

A diagram of the apoe locus with the bonding regions and the SNPs/protein changes would be helpful.

In my view, while there will be genuine "misdiagnoses" the paper below leads towards the view of blended diagnoses without clean diagnostic borders and that, I think, will underpin future genetic analyses

Shade LMP, Katsumata Y, Abner EL, Aung KZ, Claas SA, Qiao Q, Heberle BA, Brandon JA, Page ML, Hohman TJ, Mukherjee S, Mayeux RP, Farrer LA, Schellenberg GD, Haines JL, Kukull WA, Nho K, Saykin AJ, Bennett DA, Schneider JA; National Alzheimer's Coordinating Center; Ebbert MTW, Nelson PT, Fardo DW. GWAS of multiple neuropathology endophenotypes identifies new risk loci and provides insights into the genetic risk of dementia. Nat Genet. 2024 Nov;56(11):2407-2421. doi: 10.1038/s41588-024-01939-9. Epub 2024 Oct 8. PMID: 39379761; PMCID: PMC11549054.

(Remarks on code availability)

None

Reviewer #2

(Remarks to the Author)

Biobank-scale characterization of Alzheimer's disease and related dementias identifies potential disease-causing variants, risk factors, and genetic modifiers across diverse ancestries

Here, they investigated a number of large biobanks where AD/ADRD data is available. Including, All of Us, UK Biobank, 100,000 Genomes Project, Alzheimer's Disease Sequencing Project, Accelerating Medicines Partnership in Parkinson's Disease. A total of 25,001 cases and 93,542 controls. They investigated APP, PSEN1, PSEN2, TREM2, GRN, MAPT, GBA1, SNCA genes as well as APOE.

They found a higher frequency of APOE ϵ 4/ ϵ 4 carriers among individuals of African and African Admixed ancestry compared to other ancestries. At the APOE locus a disease association was found with the TOMM40:rs11556505, APOE:rs449647, 19q13.31:rs10423769, NOCT:rs13116075, CASS4:rs6024870, and LRR37A:rs2732703 variants among APOE ϵ 4 carriers across different ancestries were characterized.

Overall a very interesting log of variants in well-known genes and ApoE. Data is also presented as a browser; https://niacard.shinyapps.io/MAMBARD_browser/. Sharing these data will be important and helpful for research and clinical

assessment of patients and families.

The authors investigated a list of mainly AD genes, along with GBA, SNCA and APOE, what about the rare FTD genes such as TBK1, C9orf72 and TDP43 genes? Given the clinical diagnosis of AD in these large datasets without followup will be limited.

There are limitations as they discuss in the power of certain populations, limiting conclusions and also the lack of follow up on cases. Without follow-up clinical information many of these variants are not helpful in determining pathogenicity. In the novel and very rare variants is there a way to follow these cases up clinically?

The authors investigated ancestry from the various datasets and showed PCA plots but did they also obtain country specific African data?

In the investigated genes they present coding variants and immediate splice variants, but adding copy number and deep and possible splicing variants would also be important to help with variant interpretation within the introns of these genes as a case vs controls. Including data on ApoE variants and adding the MAPT haplotype will also be important. This will make the browser a more helpful tool in investigating coding and splicing variants.

(Remarks on code availability)

Reviewer #3

(Remarks to the Author)

This manuscript presents an interesting and useful research study. However, the main text has too many details that make it challenging to read and distracts from the message.

1) METHODS

Much of the methods section could be moved to supplementary details. I would create a single paragraph for the discovery phase, listing the generally filtering and variant selection steps with the more granular details in the supplementary methods. Additionally, some of the detail can be removed. For instance, it is not necessary to report how variants were extracted from a vcf using PLINK.

Figure 1- Label the discovery and replication cohorts. I don't think the numbers for the ancestries for each individual study are necessary, but I would like to see the overall numbers by ancestry for the discovery and replication cohort.

Figure 2- The top is unnecessary as this information is presented in Figure 1.

Figure 3- This should be a supplementary figure.

2) RESULTS

Table 1 & 2- These might be better represented in the main text as a summary figure (and the tables moved to the supplementary tables). For the entire discovery cohort, I would be interested in seeing a figure with the number of variants found per gene, the number found per ancestry, how many were known vs novel, and the number found as singletons, doubletons, etc. As well as how many were seen in the replication cohort at all, and how many replicated. After excluding known variants, do you see enrichment of pathogenic variants in cases versus controls by gene (you can compare to synonymous variants to control for population differences)? Then list the specific variants of interest in the text.

There is too much detail about which variants are seen in each individual cohort. I would compare discovery to replication.

In the section about previously reported disease-causing variants, you report the numbers in controls but not cases. While it may not be a highly pathogenic variant, are these variants associated with the disease in your cohorts?

Figure 5 and 6 probably are unnecessary. These counts/frequencies can be reported in a supplementary table. I would discuss the frequencies range of these in the text and which are common enough to be evaluated. Since these are previously reported, if you pool the protected versus risk variants do you see significant difference for cases and controls for the rarer variants?

Table 5- I would group any unknown or e1 into a single row for simplicity and group all of the cohorts.

Mention TOMM40 and APOEs proximity

Figure 6- Are any of the squares 0 in cases and 0 in controls? If so I would color those white

Figure 7- It may be more intuitive to make the size of the circle proportional to the pvalue. Do all of these populations have carriers?

(Remarks on code availability)

Version 1:

Reviewer comments:

Reviewer #1

(Remarks to the Author)

I am content

(Remarks on code availability)

Reviewer #2

(Remarks to the Author)

The authors have improved the paper with changes,

No further comments

(Remarks on code availability)

Reviewer #3

(Remarks to the Author)

Comments to authors:

I still had a question about the burden test. To clarify my question, using a gene burden test, are the rare CADD>20 novel pathogenic variants enriched in cases versus control, or even in all genes combined? Even if not significant due to lack of power, what is the OR or effect size? All EUR datasets can be combined for analysis (both discovery and replication). This gives me confidence that the CADD score is providing a reasonable estimation of pathogenicity for this analysis and that the novel variants are worth investigating for functionality or in my own cohort (The rare synonymous variant test is a control in the case that subtle differences in population structure in cases versus controls lead to more rare variants present on one population over another). The numbers present in Supplementary table 25 suggest that all nonsynonymous variants were used rather than just the predicted pathogenic ones?

(Remarks on code availability)

Response to Reviewers' comments

Thank you for the time and effort that you have devoted to the assessment of our manuscript.

IMPORTANT: Please note that we are submitting a marked and unmarked version of the revised manuscript. In the marked version, changes in text compared to the original submissions are shown in red font.

Reviewer: 1

Comment 1:

This ms represents an enormous amount of data collation and analysis which will be of value to the field (although the complexity of the analysis means that, necessarily I suspect much of the data will be difficult for researchers to find).

I have one major concern which relates to the quality of the diagnoses of "Alzheimer's disease" in general and across populations (in white US/UK populations the diagnosis is about 70% accurate... we have no idea how accurate it is in other populations). This is a serious concern which should be discussed. I note, for example, the occurrence of GRN mutations... which is an FTD gene. The recent paper on neuropath confirmed dementia cases could be cross referenced in a discussion of this issue (see bottom of review)

In my view, while there will be genuine "misdiagnoses" the paper below leads towards the view of blended diagnoses without clean diagnostic borders and that, I think, will underpin future genetic analyses

Shade LMP, Katsumata Y, Abner EL, Aung KZ, Claas SA, Qiao Q, Heberle BA, Brandon JA, Page ML, Hohman TJ, Mukherjee S, Mayeux RP, Farrer LA, Schellenberg GD, Haines JL, Kukull WA, Nho K, Saykin AJ, Bennett DA, Schneider JA; National Alzheimer's Coordinating Center; Ebbert MTW, Nelson PT, Fardo DW. GWAS of multiple neuropathology endophenotypes identifies new risk loci and provides insights into the genetic risk of dementia. Nat Genet. 2024 Nov;56(11):2407-2421. doi: 10.1038/s41588-024-01939-9. Epub 2024 Oct 8. PMID: 39379761; PMCID: PMC11549054.

Response:

We appreciate Reviewer 1 for highlighting the critical issue of diagnostic accuracy for Alzheimer's disease and dementia across diverse populations and biobanks. Indeed, when utilizing biobank-related data, we must contend with the inherent heterogeneity and potential biases arising from varying diagnostic sources, the absence of standardized clinical criteria, and inconsistent

methodologies. We acknowledge and have addressed this limitation in the discussion section, emphasizing that diagnostic accuracy remains a challenge for two key reasons: first, the potential for misdiagnosis in approximately 30% of cases among white populations; and second, the lack of applicability of current diagnostic criteria across populations. For instance, MoCA thresholds for Europeans differ from those for African Americans, largely due to differences in education levels. Additionally, we have referenced the literature provided by the reviewer, where we highlight that reliance on diagnoses without clear boundaries will likely underpin future genetic analyses. Even under optimal conditions, some cases may remain ambiguous, and mixed or atypical dementias may be mistakenly classified under AD/ADRD categories, further complicating genetic investigations.

In addition to the discussion, we also aimed to provide the most accurate information regarding the specific criteria each biobank uses to define AD/ADRD cases, ensuring that the reader is aware of this important variability in the Methods section.

- **Methods section, Paragraph 19, lines 739-754:**

Diagnostic Criteria across Biobanks

AoU: Diagnoses are primarily derived from electronic health records (EHRs) and linked to ICD-9/ICD-10 codes.

UKB: Diagnoses are obtained through linkage to national health registries, including hospital episode statistics and death certificates. UK Biobank's algorithmically-defined outcomes are the primary source for identifying AD/ADRD cases.

100KGP: This project focuses on genomic data, with clinical diagnoses often drawn from medical records provided by referring clinicians. Diagnoses originate from hospitals in collaboration with Genomics England (GE), where expert neurologists evaluate patients and submit diagnostic data to GE.

ADSP: Diagnoses are often based on standardized research protocols such as NINCDS-ADRDA or DSM-IV/DSM-5 criteria. Many cases include neuropathological confirmation or biomarker evidence (e.g., CSF amyloid/tau levels).

AMP-PD: While primarily focused on PD, cases of mixed dementia are identified through clinical evaluations and medical records. Cases overlapping with AD/ADRD are typically based on clinical diagnoses supported by neurological assessments.

In some databases, additional measures are taken to confirm diagnoses. For example, in the UK Biobank, they use algorithmically defined outcomes, which take extra steps to ensure accurate diagnoses rather than relying solely on preliminary diagnoses. These algorithmically defined outcomes include probable cases of selected health conditions, determined by combining coded

information from UK Biobank's baseline assessment data (e.g., self-reported medical conditions, operations, and medications) with linked data from hospital admissions, diagnoses, procedures, and death registries. The purpose of these derived data fields is to help researchers include health-related outcomes in their analyses without manually selecting diagnostic or procedural codes and combining data sources themselves. These algorithms were developed by the UK Biobank Outcome Adjudication group to improve diagnostic reliability.

In the ADSP dataset, we included only diagnoses of "definitive AD" in our analysis, and even "probable AD" cases were excluded.

We have addressed these two valuable points and expanded our limitations section accordingly:

Methods, Paragraph 9, lines 644-648:” The AD cohort was defined by the UKB field ID 42020, using diagnoses according to the UKB's algorithmically defined outcomes v2.0 (<https://biobank.ndph.ox.ac.uk/ukb/refer.cgi?id=460>). The related dementia cohort was defined by the UKB field ID 42018, using the UKB's algorithmically defined “Dementia” classification, with the added step of excluding any individuals in the aforementioned AD cohort. “

Methods, Paragraph 16, line 713-714:” Only samples labeled as definite AD or control were included in this analysis.”

Discussion, Paragraph 10, lines 548-554: “One critical concern is the potential for misdiagnosis across different populations, which can result from overlapping clinical presentations and the inclusion of individuals with mixed or alternative dementia etiologies. This issue highlights the challenges in delineating clear diagnostic borders in dementia research, underscoring the need for neuropathologically confirmed cohorts and reinforcing the importance of blended diagnosis in future genetic analyses⁷². Additionally, inter-ancestry differences in diagnostic criteria exist. For instance, optimal cutoffs for the Montreal Cognitive Assessment require stratification by race, ethnicity, and education⁷³.”

Comment 2:

A diagram of the apoe locus with the bonding regions and the SNPs/protein changes would be helpful.

Response:

Thank you for your excellent comment. We have added a new figure to the manuscript (Figure 3), which summarizes genetic results for all the genes.

Reviewer: 2

Comment 1:

Biobank-scale characterization of Alzheimer's disease and related dementias identifies potential disease-causing variants, risk factors, and genetic modifiers across diverse ancestries

Here, they investigated a number of large biobanks where AD/ADRD data is available. Including, All of Us, UK Biobank, 100,000 Genomes Project, Alzheimer's Disease Sequencing Project, Accelerating Medicines Partnership in Parkinson's Disease. A total of 25,001 cases and 93,542 controls. They investigated APP, PSEN1, PSEN2, TREM2, GRN, MAPT, GBA1, SNCA genes as well as APOE.

They found a higher frequency of APOE $\epsilon 4/\epsilon 4$ carriers among individuals of African and African Admixed ancestry compared to other ancestries. At the APOE locus a disease association was found with the TOMM40:rs11556505, APOE:rs449647, 19q13.31:rs10423769, NOCT:rs13116075, CASS4:rs6024870, and LRRC37A:rs2732703 variants among APOE $\epsilon 4$ carriers across different ancestries were characterized.

Overall a very interesting log of variants in well-known genes and ApoE. Data is also presented as a browser; https://niacard.shinyapps.io/MAMBARD_browser/. Sharing these data will be important and helpful for research and clinical assessment of patients and families.

Response:

Thank you for the positive response. We are delighted that you found the manuscript, data presentation, and browser useful. We hope that sharing these data will contribute to future research efforts and clinical assessments, ultimately benefiting patients and their families.

Comment 2:

The authors investigated a list of mainly AD genes, along with GBA, SNCA and APOE, what about the rare FTD genes such as TBK1, C9orf72 and TDP43 genes? Given the clinical diagnosis of AD in these large datasets without followup will be limited.

There are limitations as they discuss in the power of certain populations, limiting conclusions and also the lack of follow up on cases. Without follow-up clinical information many of these variants are not helpful in determining pathogenicity. In the novel and very rare variants is there a way to follow these cases up clinically?

Response:

Thank you for your excellent points. As suggested, data for *TBK1*, *TDP43 (TARDBP)*, and *APOE* have been incorporated into the manuscript.

See Tables 1- 4, Supplementary Tables 1-6, and Result section.

The current study did not focus on the *C9ORF72* repeat expansion due to its complex nature. Analyzing repeat expansions requires access to CRAM/BAM files, and current approaches on short-read WGS data do not accurately detect these. Processing the associated alignment files across all biobanks is currently infeasible due to significant storage and cost constraints. Nevertheless, we acknowledge its significance and plan to address it in a dedicated future project. We have included this limitation as follows:

Discussion, Paragraph 10, lines 559-562: “In addition, the current study did not focus on the *C9ORF72* repeat expansion due to its complex nature, as current approaches using short-read WGS data do not accurately detect it. Future studies should address this limitation.”

We completely agree that the lack of clinical follow-up across biobanks limits the ability to fully assess the pathogenicity of certain variants. We hope the data browser we have developed acts as an interactive tool for researchers to report additional carriers and track their own cohorts with clinical follow-ups outside of the biobank space. We have added this important limitation as follows:

Discussion, Paragraph 10, lines 541-546: “A major limitation is the underrepresentation of certain populations and the lack of clinical follow-up on cases, both of which lead to underpowered datasets and limit the possibility of drawing firm conclusions. It is our hope that the interactive tool we have developed will help the research community further investigate these mutations in clinical and longitudinally characterized cohorts in a global context.”

Comment 3:

The authors investigated ancestry from the various datasets and showed PCA plots but did they also obtain country specific African data?

Response:

Thank you for your insightful question. According to the ancestry prediction estimates from GenoTools, our cohorts included samples from 11 general ancestries. Currently, our ability to genetically predict subpopulations is limited due to the lack of publicly available reference panels. As a result, we were unable to provide precise origin information for many of these samples beyond the general predictions from GenoTools. The African ancestry data used in our analysis

was aggregated across populations rather than being broken down by individual countries or sub-populations. In an attempt to further explore this, the figure below shows our data clustered with African subpopulations according to the 1000 Genomes Project, highlighting the challenges and inaccuracies in predicting genetic subpopulations. We have discussed this limitation as follows:

Discussion, Paragraph 10, lines 554-556: “Other relevant considerations include the differing exclusion criteria for controls across cohorts and the limitations of genetically predicted subpopulations, which may affect the comparability of results.”

Comment 4:

In the investigated genes they present coding variants and immediate splice variants, but adding copy number and deep and possible splicing variants would also be important to help with variant interpretation within the introns of these genes as a case vs controls. Including data on ApoE variants and adding the MAPT haplotype will also be important. This will make the browser a more helpful tool in investigating coding and splicing variants.

Response:

Thank you for your suggestions. We completely agree that deep intronic and copy number variants (CNVs) are very important. However, using short-read WGS to analyze CNVs may introduce

some bias. We plan to conduct a long-read sequencing study once biobanks release these data. In addition, as we mentioned before, analyzing CNVs requires specialized computational resources, and processing the associated alignment files across all biobanks is currently infeasible due to significant storage and cost constraints. This is a pilot study focused on protein-altering and splicing variants, and our next immediate step in a future study will be to conduct a multi-ancestry AD meta-analysis examining common intronic variation.

We truly appreciate your comments on *APOE* and *MAPT* haplotypes. Results on *APOE* variants have been added to the manuscript as recommended, and the characterization of the *MAPT* haplotype is part of an ongoing collaborative project, which will be considered as a separate publication.

Reviewer: 3

Comment 1:

This manuscript presents an interesting and useful research study. However, the main text has too many details that make it challenging to read and distracts from the message.

1) METHODS

Much of the methods section could be moved to supplementary details. I would create a single paragraph for the discovery phase, listing the generally filtering and variant selection steps with the more granular details in the supplementary methods. Additionally, some of the detail can be removed. For instance, it is not necessary to report how variants were extracted from a vcf using PLINK.

Response:

We appreciate your feedback on the methods section. While we totally agree and understand the suggestion to streamline the methods by moving granular details as part of the supplementary section, the senior editor specifically has advised us to retain all the details in the main text to ensure transparency and reproducibility. To align with the journal's preferences, we have kept the methods section intact so they will be able to further restructure it if needed according to the journal requirements.

Comment 2:

Figure 1- Label the discovery and replication cohorts. I don't think the numbers for the ancestries for each individual study are necessary, but I would like to see the overall numbers by ancestry for the discovery and replication cohort.

Response:

Thank you. As you can see, we have separated the discovery phase on the right and the replication phase on the left of the figure. Additionally, we have added a new row for the overall numbers by ancestry for the discovery and replication cohorts.

Comment 3:

Figure 2- The top is unnecessary as this information is presented in Figure 1.

Response:

Thank you. The figure has been updated to remove the redundant information. We've kept the map to highlight that the data from the biobanks represent two distinct major regional areas.

Comment 4:

Figure 3- This should be a supplementary figure.

Response:

Thank you. This is now Supplementary Figure 1.

Comment 5:

2) RESULTS

Table 1 & 2- These might be better represented in the main text as a summary figure (and the tables moved to the supplementary tables). For the entire discovery cohort, I would be interested in seeing a figure with the number of variants found per gene, the number found per ancestry, how many were known vs novel, and the number found as singletons, doubletons, etc. As well as how many were seen in the replication cohort at all, and how many replicated.

Response:

Thank you. Figure 3, containing all the requested information, has been added to the article.

Comment 6:

After excluding known variants, do you see enrichment of pathogenic variants in cases versus controls by gene (you can compare to synonymous variants to control for population differences)? Then list the specific variants of interest in the text

Response:

Thank you for this suggestion. We performed a burden analysis to determine whether there was an enrichment of pathogenic variants in cases compared to controls by gene, after excluding known variants. This analysis also controlled for synonymous variants to account for population differences. We have now added the methods and results of this analysis to the manuscript, as follows:

Methods, Paragraph 30, lines 828-831: “A burden analysis for rare variants ($MAF \leq 0.01$) in the 11 genes studied was performed using the European ancestry group of the ADSP dataset. This analysis was conducted by variant/functional category: the first analysis included only synonymous variants, while the second analysis excluded synonymous variants, as well as known coding and splicing variants listed in Table 1.”

Results, Paragraph 37, lines 440-443: “We performed a burden analysis to determine whether there was an enrichment of pathogenic variants in cases compared to controls by gene, after excluding known variants. Burden testing, adjusted for sex, age, and PCs using SKAT-O, revealed an enrichment of rare variants in *GBAI* ($P = 0.023$) (**Supplementary Tables 25 and 26**).”

Comment 7:

There is too much detail about which variants are seen in each individual cohort. I would compare discovery to replication.

Response:

Thank you for your feedback. In the Results section (lines 127–161), we present our findings from the discovery phase, while lines 163–229 focus on the replication phase. Although we aimed to avoid excessive detail, we included a comparison of variants across ancestries to guide readers through the Supplementary Tables and highlight potential ethnicity-driven differences. We made sure to stay within the journal's word limit while providing this information.

Comment 8:

In the section about previously reported disease-causing variants, you report the numbers in controls but not cases. While it may not be a highly pathogenic variant, are these variants associated with the disease in your cohorts?

Response:

Thank you. The frequency and number of carriers for each variant in cases and controls have been added as part of Supplementary Table 4.

This sentence was added to the article: Result, Paragraph 14, lines 262-264: "The frequency and number of carriers for each variant in cases and controls are reported in **Supplementary Table 4.**"

Comment 9:

Figure 5 and 6 probably are unnecessary. These counts/frequencies can be reported in a supplementary table. I would discuss the frequencies range of these in the text and which are common enough to be evaluated. Since these are previously reported, if you pool the protective versus risk variants do you see significant differences for cases and controls for the rarer variants?

Response:

Thank you for your suggestions. The counts and frequencies shown in Figures 5 and 6 are now reported in Supplementary Tables 8–13 and 14–23, respectively. Based on your suggestions, we have also moved Figures 5 and 6 to the Supplementary Material. They are now Supplementary Figures 4 and 5.

To address the second point, we performed a polygenic risk score analysis using summary statistics from Kunkle et al., 2019. We identified individuals with the highest genetic risk burden (top 25th percentile), which included both cases and controls. Next, we performed a logistic regression analysis for protective/disease-modifying variants using two separate models—one adjusted for Z score, sex, age, and PCs, and the other adjusted for Z score, *APOE* status, sex, age, and PCs. We then assessed the potential enrichment of these variants in cases versus controls. The results were added to the manuscript and Supplementary Table 24.

We expanded our manuscript as follows:

Methods, Paragraph 29, lines 822-827: "PRS was calculated using the summary statistics from Kunkle 2019⁸⁹ for European ancestry samples in ADSP data. Individuals with the highest genetic risk burden (top 25th percentile), including both cases and controls, were then subsetted. A logistic regression analysis of all 21 protective/disease-modifying variants was performed using two separate models: one adjusted for Z score, sex, age, and PCs, and the other adjusted for Z score, *APOE* status, sex, age, and PCs, applied to the selected individuals."

Results, Paragraph 36, lines 433-438:” To assess the potential enrichment of protective or disease-modifying variants in cases versus controls, we conducted logistic regression on individuals with the highest genetic risk burden based on the PRS, including 1,997 cases and 559 controls. Results are presented in **Supplementary Table 24**. The analysis revealed no significant differences between cases and controls for these variants and indicated no interaction between these variants and PRS to modify disease penetrance. This analysis may be underpowered due to the low frequency of these variants.”

Comment 10:

Table 5- I would group any unknown or e1 into a single row for simplicity and group all of the cohorts.

Mention TOMM40 and APOEs proximity

Response:

Thank you for this suggestion. For simplicity, we moved Table 5 to the Supplementary Material (this is now Supplementary Table 7), and the combined results across all datasets for AD, related dementias, and controls for all genotypes are shown in Supplementary Figure 3. Only data for *APOE* e4/e4 are shown in Figure 4.

We have also clarified *TOMM40* and *APOE* proximity as follows:

Discussion, Paragraph 8, lines 520–521: “Of note, *APOE* and *TOMM40* are located in close proximity on chromosome 19q13.32, with *TOMM40* positioned immediately upstream of *APOE*.”

Comment 11:

Figure 6- Are any of the squares 0 in cases and 0 in controls? If so I would color those white

Response:

Thank you so much for the excellent suggestion. All 0 values in cases and controls were colored white, and this sentence was added to the Figure’s legend: '0 values in cases and controls are shown in white'. This is now Supplementary Figure 4.

Comment 12:

Figure 7- It may be more intuitive to make the size of the circle proportional to the pvalue. Do all of these populations have carriers?

Response:

Thank you. Rather than changing the size of the circles, we decided to follow a more visually friendly approach. P-values < 0.05 are shown with asterisks, and the following sentence was added to the legend: '* Indicates p-value < 0.05 .' This is now Supplementary Figure 5. In response to your question - Yes, all these populations contain carriers.

REVIEWER COMMENTS

Reviewer #1 (Remarks to the Author):

I am content

Reviewer #2 (Remarks to the Author):

The authors have improved the paper with changes,
No further comments

Reviewer #3 (Remarks to the Author):

Comments to authors:

I still had a question about the burden test. To clarify my question, using a gene burden test, are the rare CADD>20 novel pathogenic variants enriched in cases versus control, or even in all genes combined? Even if not significant due to lack of power, what is the OR or effect size? All EUR datasets can be combined for analysis (both discovery and replication). This gives me confidence that the CADD score is providing a reasonable estimation of pathogenicity for this analysis and that the novel variants are worth investigating for functionality or in my own cohort (The rare synonymous variant test is a control in the case that subtle differences in population structure in cases versus controls lead to more rare variants present on one population over another). The numbers present in Supplementary table 25 suggest that all nonsynonymous variants were used rather than just the predicted pathogenic ones?

Response: Thank you for your comments. We would like to clarify our criteria in response.

In this study, variants were prioritized based on their **absence in controls (Results, lines 137, 149, and 160) and a CADD score >20 (Table 1 and 2). Consequently, after prioritization, the frequency of these variants in the controls is zero, making further burden testing irrelevant for these specific variants.**